# EXPECTED PERTURBATION SCORE FOR ADVERSARIAL DETECTION

## ABSTRACT

Adversarial detection aims to determine whether a given sample is an adversarial one based on the discrepancy between natural and adversarial distributions. Unfortunately, estimating or comparing two data distributions is extremely difficult, especially in the high-dimension space. Recently, the *gradient* of log probability density (a.k.a., score) w.r.t. the sample is used as an alternative statistic to compute. However, we find that the score is sensitive in identifying adversarial samples due to insufficient information with one sample only. In this paper, we propose a new statistic called *expected perturbation score* (EPS), which is essentially the expected score of a sample after various perturbations. Specifically, to obtain adequate information regarding one sample, we can perturb it by adding various noises to capture its multi-view observations. We theoretically prove that EPS is a proper statistic to compute the discrepancy between two samples under mild conditions. In practice, we can use a pre-trained diffusion model to estimate EPS for each sample. Last, we propose an EPS-based adversarial detection (EPS-AD) method, in which we develop EPS-based *maximum mean discrepancy* (MMD) as a metric to measure the discrepancy between the test sample and natural samples. To verify the validity of our proposed method, we also prove that the EPS-based MMD between natural and adversarial samples is larger than that among natural samples. Empirical studies on CIFAR-10 and ImageNet across different network architectures including ResNet, WideResNet, and ViT show the superior adversarial detection performance of EPS-AD compared to existing methods.

## 1 INTRODUCTION

Deep neural networks (DNNs) are known to be sensitive to adversarial samples that are generated by adding imperceptible perturbations to the input but may mislead the model to make unexpected predictions (Szegedy et al., 2014; Goodfellow et al., 2015). Adversarial samples threaten widespread machine learning systems (Li & Vorobeychik, 2014; Ozbulak et al., 2019), which raises an urgent requirement for advanced techniques to improve the robustness of models. Among them, *adversarial training* introduces adversarial data into training to improve the robustness of models but suffers from significant performance degradation and high computational complexity (Sriramanan et al., 2021; Laidlaw et al., 2021; Wong et al., 2020); *adversarial purification* relies on generative models to purify adversarial data before classification, which still has to compromise on unsatisfactory natural and adversarial accuracy (Shi et al., 2021; Yoon et al., 2021; Nie et al., 2022).

In contrast, another class of defense methods, called *adversarial detection*, could be achieved by detecting and rejecting adversarial examples, which are friendly to existed machine learning systems due to the lossless natural accuracy, and can help to identify security-compromised input sources (Abusnaina et al., 2021). Adversarial detection aims to tell whether a test sample is an adversarial sample, for which the key is to find the discrepancy between the adversarial and natural distributions. However, existing adversarial detection approaches primarily train a tailored detector for specific attacks (Feinman et al., 2017; Ma et al., 2018; Lee et al., 2018) or for a specific classifier (Deng et al., 2021), which largely overlook modeling the adversarial and natural distributions, resulting in their limited performance against unseen attacks or transferable attacks.

Unfortunately, it is non-trivial to estimate or compare two data distributions, especially in the high-dimension space (*e.g.*, image-based space). One alternative approach is to estimate the *gradient*

of log probability density with respect to the sample, *i.e.*, score. This statistic has emerged as a powerful means for adversarial defense (Yoon et al., 2021; Nie et al., 2022) and diffusion models (Song & Ermon, 2019; Song et al., 2021; Kingma et al., 2021; Huang et al., 2021). However, how to effectively exploit the score function for adversarial detection is not well studied.

Recently, Yoon et al. (2021) purify adversarial samples by gradually removing the adversarial noise from the (attacked) samples with the score function for adversarial defense. During the purification process, they employ the *norm* of scores (between being-purified adversarial samples and natural samples) to set a threshold for determining which timestep to stop purifying. They empirically find that natural samples usually have lower score norms than adversarial samples across purification timesteps. Intuitively, the score could represent the momentum of the sample towards the high-density areas of natural data (Song & Ermon, 2019). From this point of view, a lower score norm indicates the sample is closer to the high-density areas of natural data, *i.e.*, a higher probability of the sample following the natural distribution. To further understand this, we demonstrate the score norms of natural samples and adversarial samples at different purification timesteps. According to Figure 1, most natural samples have lower score norms than adversarial samples at the same timestep, but they

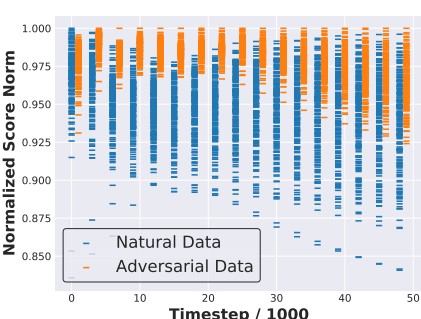

Figure 1: An illustration of score norms of 200 random sampled natural images and adversarial images on ImageNet, in which most natural images have lower score norms than adversarial images at the same timestep but are very sensitive to the timesteps due to the significant overlap.

are very sensitive to the timesteps due to the significant overlap across all timesteps. This suggests that the score of one sample is useful but not effective enough in identifying the adversarial samples.

In this paper, we propose a new statistic called *expected perturbation score* (EPS), which represents the expected score after multiple perturbations of a given sample. In EPS, we consider multiple levels of noise perturbations to diversify one sample, allowing us to capture multi-view observations of one sample and thus extract adequate information from the data. Our theoretical analysis shows EPS is a valid statistic in distinguishing between natural and adversarial samples under mild conditions. Thus, we propose an *EPS-based adversarial detection* method (EPS-AD) for adversarial detection, as illustrated in Figure 2. Specifically, given a pre-trained score-based diffusion model, EPS-AD consists of three steps: 1) we simultaneously add multiple perturbations to a set of natural images and an upcoming image following the forward diffusion process with a time step $T^*$; 2) we obtain their EPSs via the score model; 3) we adopt the *maximum mean discrepancy* (MMD) between any test sample and natural samples relying on EPS.

We provide both empirical and theoretical analyses to demonstrate the effectiveness of EPS-AD. Empirically, we achieve superior performance on both CIFAR-10 and ImageNet across many network architectures including ResNet, WideResNet and ViT compared to existing methods. Especially for the extremely low attack intensity $\epsilon = 1/255$ on ImageNet, we achieve the area under the receiver operating characteristic (AUROC) of $+95\%$ against 12 attacks over ResNet-50. Theoretically, we prove that the MMD between EPSs of the natural samples is smaller than that between natural and adversarial samples.

We summarize our main contributions as follows:

- We propose a new and reliable statistic called *expected perturbation score* (EPS) to capture sufficient information regarding a sample from its multi-view observations by adding various perturbations. We theoretically prove that EPS is a proper statistic to compute the discrepancy between two distributions under mild conditions.

- Relying on the proposed EPS, we exploit the *maximum mean discrepancy* (MMD) as a metric to measure the discrepancy between the test sample and natural samples and then develop a novel adversarial detection method called EPS-AD. We theoretically show that the EPS-based MMD between natural and adversarial samples is larger than that among natural samples, which verifies the validity of the proposed adversarial detection method.

## 2 PRELIMINARIES

We start by recalling the setting of score-based continuous-time diffusion models, and then present the concept of *maximum mean discrepancy*.

**Adversarial data generation.** Given a well-trained classifier $\hat{f}$ on a data set $D\{(\mathbf{x}_i, l_i)\}_{i=1}^n$ with $\mathbf{x}_i$ being a sample from the input space $\mathcal{X}$ and $l_i$ being its ground-truth label defined in a label set $\mathcal{C} = \{1, \cdots, C\}$, adversarial data $\hat{\mathbf{x}}$ regarding $\mathbf{x}$ with perturbation $\epsilon$ is given by

$$\hat{\mathbf{x}} = \underset{\hat{\mathbf{x}} \in \mathcal{B}(\mathbf{x}, \epsilon)}{\arg\max} \ell\left(\hat{f}(\hat{\mathbf{x}}), l\right), \tag{1}$$

where $\mathcal{B}(\mathbf{x}, \epsilon) = \{\mathbf{x}' \in \mathcal{X} \mid d(\mathbf{x}, \mathbf{x}') \le \epsilon\}$, $d$ is some distance (*e.g.*, $\ell_2$ or $\ell_\infty$ distance), and $\ell$ is some loss function. For simplicity, we denote $\hat{\mathbf{x}} = \mathbf{x} + \epsilon$ as the adversarial data regarding $\mathbf{x}$.

**Continuous-time diffusion models.** Following Song et al. (2021), let $p(\mathbf{x})$ be the unknown data distribution. Diffusion models firstly construct a forward diffusion process $\{\mathbf{x}_t\}_{t=0}^{T_{\text{diff}}}$ indexed by a continuous time variable $t \in [0, T_{\text{diff}}]$, which can be modeled by a stochastic differential equation (SDE) with positive time increments:

$$d\mathbf{x} = \mathbf{f}(\mathbf{x}, t)dt + g(t)d\mathbf{w}, \tag{2}$$

where $\mathbf{x}_0 := \mathbf{x} \sim p(\mathbf{x})$, $\mathbf{f}(\cdot, t) : \mathbb{R}^d \to \mathbb{R}^d$ is a vector-valued function, $g(\cdot) : \mathbb{R} \to \mathbb{R}$ is a scalar function that is independent on $\mathbf{x}$, and $\mathbf{w}$ is a standard Wiener process.

Let $p_t(\mathbf{x})$ be the the marginal distribution of $\mathbf{x}_t$ with $p_0(\mathbf{x}) = p(\mathbf{x})$, if $\mathbf{f}(\mathbf{x}, t)$ and $g(t)$ are designed well such that $p_T(\mathbf{x}) \approx \mathcal{N}(\mathbf{0}, \mathbf{I}_d)$, by reversing the diffusion process from $t = T_{\text{diff}}$ to $t = 0$, we can reconstruct samples $\mathbf{x}_0 \sim p_0(\mathbf{x})$. The reverse process is given by the reverse-time SDE:

$$d\mathbf{x} = \left[\mathbf{f}(\mathbf{x}, t) - g(t)^2 \nabla_{\mathbf{x}} \log p_t(\mathbf{x})\right] dt + g(t)d\bar{\mathbf{w}}, \tag{3}$$

where $\bar{\mathbf{w}}$ is a standard reverse-time Wiener process and $dt$ is an infinitesimal negative time step. Throughout the paper, we consider the VP-SDE following the setting of Song et al. (2021), where $\mathbf{f}(\mathbf{x}, t) := -\frac{1}{2}\beta(t)\mathbf{x}$ and $g(t) := \sqrt{\beta(t)}$ with $\beta(t)$ being the linear noise schedule, *i.e.*, $\beta(t) := \beta_{\min} + (\beta_{\max} - \beta_{\min}) t/T_{\text{diff}}$ for $t \in [0, T_{\text{diff}}]$.

Reconstructing samples from the Gaussian distribution requires the score of the marginal distribution, *i.e.*, $\nabla_{\mathbf{x}} \log p_t(\mathbf{x})$ in the reverse process Eq. (3). To estimate the score function $\nabla_{\mathbf{x}} \log p_t(\mathbf{x})$, one effective solution is to train a score model $\mathbf{s}_{\boldsymbol{\theta}}(\mathbf{x}, t)$ on samples with score matching (Hyvärinen & Dayan, 2005; Song & Ermon, 2019; Vincent, 2011). The training objective function is:

$$\boldsymbol{\theta}^* = \min_{\boldsymbol{\theta}} \mathbb{E}_t \left\{ \lambda(t) \mathbb{E}_{\mathbf{x}_0 \sim p_0(\mathbf{x}_0)} \mathbb{E}_{\mathbf{x}_t \sim p_{0t}(\mathbf{x}_t | \mathbf{x}_0)} \|s_{\boldsymbol{\theta}}(\mathbf{x}_t, t) - \nabla_{\mathbf{x}_t} \log p_{0t}(\mathbf{x}_t \mid \mathbf{x}_0)\|_2^2 \right\}, \tag{4}$$

where $\lambda(t)$ is a positive weighting function, and $p_{0t}(\mathbf{x}_t \mid \mathbf{x}_0)$ is a transition kernel from $\mathbf{x}_0$ to $\mathbf{x}_t$, which can be derived by the forward SDE in Eq. (2).

**Maximum mean discrepancy.** Following Gretton et al. (2012); Borgwardt et al. (2006), let $\mathcal{X} \subset \mathbb{R}^d$ be a separable metric space and $p, q$ be Borel probability measures on $\mathcal{X}$. Given two independent identically distributed (*IID*) observations $S_X = \{\mathbf{x}^{(i)}\}_{i=1}^n$ and $S_Y = \{\mathbf{y}^{(i)}\}_{i=1}^m$ from distributions $p$ and $q$, respectively, *maximum mean discrepancy* (MMD) aims to measure the closeness between these two distributions, which is defined as:

$$\text{MMD}(p, q; \mathcal{F}) := \sup_{f \in \mathcal{F}} |\mathbb{E}[f(X)] - \mathbb{E}[f(Y)]|, \tag{5}$$

where $\mathcal{F}$ is a class of functions $f : \mathcal{X} \to \mathbb{R}$, and $X \sim p, Y \sim q$ are two random variables. To better study the richness of the MMD function class $\mathcal{F}$, Borgwardt et al. (2006) propose to choose $\mathcal{F}$ to be the unit ball in a universal reproducing kernel Hilbert space and obtain the following kernel-based MMD,

$$\begin{aligned}\text{MMD}(p, q; \mathcal{H}_k) &:= \sup_{f \in \mathcal{H}, \|f\|_{\mathcal{H}_k} \le 1} |\mathbb{E}[f(X)] - \mathbb{E}[f(Y)]| \\ &= \|\mu_p - \mu_q\|_{\mathcal{H}_k} = \sqrt{\mathbb{E}[k(X, X') + k(Y, Y') - 2k(X, Y)]},\end{aligned} \tag{6}$$

where $k : \mathcal{X} \times \mathcal{X} \to \mathbb{R}$ is the kernel of a reproducing kernel Hilbert space $\mathcal{H}_k$, $\mu_p := \mathbb{E}[k(\cdot, X)]$ and $\mu_q := \mathbb{E}[k(\cdot, Y)]$ are the kernel mean embeddings of $p$ and $q$, respectively (Gretton et al., 2012; Borgwardt et al., 2006; Jitkrittum et al., 2017; Liu et al., 2020; Gao et al., 2021).

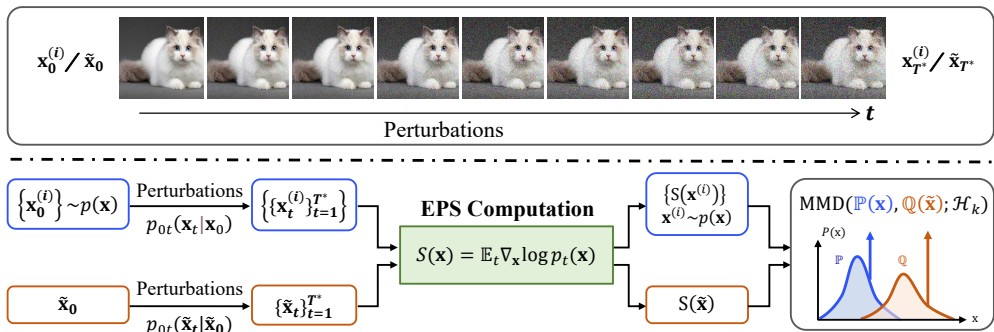

Figure 2: Overview of the proposed EPS-AD. EPS denotes the expected score after multiple perturbations of a sample using a pre-trained score model. Specifically, we simultaneously add perturbations to a set of natural images $\{\mathbf{x}_0^{(i)}\}$ and a test image $\tilde{\mathbf{x}}_0$ following the diffusion process with a time step $T^*$ to get perturbed images, from which we obtain their EPSs $S(\mathbf{x})$ via the score model and calculate the MMD between EPS of the test sample and EPSs of natural samples.

## 3 PROPOSED METHOD

In this section, we first provide the problem definition of *adversarial detection* in Section 3.1. Then we present the definition of *expected perturbation score* (EPS) and provide the theoretical analysis that EPS is able to distinguish between natural and adversarial data well if natural data are from Gaussian distributions (Section 3.2). Motivated by this, we develop a new adversarial detection method, called EPS-AD, as shown in Figure 2. Particularly, we estimate the EPS for each sample using a pre-trained score-based diffusion model and then adopt the *maximum mean discrepancy* (MMD) between the test sample and natural samples relying on EPS as a characteristic of the test sample (Section 3.3). Moreover, we provide theoretical analysis that the MMD between EPSs of the natural samples is smaller than that between natural and adversarial samples. (Section 3.4).

### 3.1 PROBLEM SETTING

In this paper, we aim to address the following adversarial detection problem.

**Problem 1.** *(Adversarial detection) Let $\mathcal{X} \subset \mathbb{R}^d$ be a separable metric space and $p$ be a Borel probability measure on $\mathcal{X}$, and IID observations $S_X = \{\mathbf{x}^{(i)}\}_{i=1}^n$ from the distribution $p$ and a ground-truth labeling mapping $f(\cdot) := \mathbb{R}^d \to \mathcal{C}$ with $\mathcal{C} = \{1, \dots, C\}$ being a label set. Assuming that the attacker has access to some well-trained classifier $\hat{f}$ on $S_X$ and IID observations $S'_X$ from the distribution $p$, we wish to know whether each new sample in $S_Y = \{\mathbf{y}^{(i)}\}_{i=1}^m$ that are crafted with $S'_X$ is following the distribution $p$.*

Note that the definition of *adversarial detection* in Problem 1 is different from that in two-sample test (Grosse et al., 2017). Particularly, Problem 1 aims to determine whether each example in $S_Y$ is sampled from the distribution $p$, while two-sample test aims to tell if two populations $S_X$ and $S_Y$ come from the same distribution, which focuses on the closeness between two populations.

### 3.2 EXPECTED PERTURBATION SCORE

As aforementioned, the score of one sample, *e.g.*, $\nabla_{\mathbf{x}} \log p(\mathbf{x})$, is ineffective enough in identifying the adversarial samples. To capture more information from one sample, we propose a new statistic *expected perturbation score* (EPS), which is formulated as bellow.

**Definition 1.** *(Expected perturbation score) Let $\mathcal{X} \subset \mathbb{R}^d$ be a separable metric space and $p$ be Borel probability measure on $\mathcal{X}$. Given a perturbation process transition distribution $p_{0t}(\mathbf{x}_t \mid \mathbf{x}_0)$, the expected perturbation score (EPS) of a sample $\mathbf{x}$ is given by:*

$$S(\mathbf{x}) = \mathbb{E}_{t \sim u(0,T)} \nabla_{\mathbf{x}} \log p_t(\mathbf{x}), \tag{7}$$

*where $p_t(\mathbf{x})$ is the marginal probability distribution of $\mathbf{x}_t$, $\mathbf{x}_0 := \mathbf{x}$ is the initialized sample at $t = 0$, and $T$ is the last perturbation time step.*

Note that the perturbation process transition distribution $p_{0t}(\mathbf{x}_t \mid \mathbf{x}_0)$ can be any distribution, such as the commonly used Gaussian distribution or uniform distribution. Moreover, we consider multiple levels of noise in the definition of EPS, with the aim of diversifying the single sample, which enables us to capture multi-view observation data and thus fully exploit more information from the data.

Built upon Definition 1, we derive a following theorem to give a closer look on EPS $S(x)$ for the natural data and adversarial data when the perturbation process transition distribution $p_{0t}(\mathbf{x}_t \mid \mathbf{x}_0)$ and $p(x)$ are from Gaussian distributions.

**Theorem 1.** *Assuming that the distribution of natural data $p(\mathbf{x}) = \mathcal{N}(\boldsymbol{\mu}_\mathbf{x}, \sigma_\mathbf{x}^2 \mathbf{I})$, given a perturbation transition kernel $p_{0t}(\mathbf{x}_t \mid \mathbf{x}_0) = \mathcal{N}(\mathbf{x}_t; \gamma_t \mathbf{x}_0, \sigma_t^2 \mathbf{I})$ with $\gamma_t$ and $\sigma_t$ being the time-dependent noise schedule, then the following three conclusions hold for $S(\mathbf{x})$:*
*1) For $\forall\, \mathbf{x} \sim p(\mathbf{x})$, $S(\mathbf{x}) \sim \mathcal{N}(\mathbf{0}, \sigma_S^2 \mathbf{I})$;*
*2) For $\forall\, \mathbf{y} \sim p(\mathbf{x})$, adversarial sample $\hat{\mathbf{y}} = \mathbf{y} + \epsilon$ with perturbation $\epsilon$, $S(\hat{\mathbf{y}}) \sim \mathcal{N}(-\boldsymbol{\mu}_S, \sigma_S^2 \mathbf{I})$;*
*3) For $\forall\, \mathbf{x}, \mathbf{y} \sim p(\mathbf{x})$, adversarial sample $\hat{\mathbf{y}} = \mathbf{y} + \epsilon$, we have*

$$S(\mathbf{x}) - S(\mathbf{y}) \xrightarrow{d} \mathcal{N}(\mathbf{0}, 2\sigma_S^2 \mathbf{I}); \tag{8}$$

$$S(\mathbf{x}) - S(\hat{\mathbf{y}}) \xrightarrow{d} \mathcal{N}(\boldsymbol{\mu}_S, 2\sigma_S^2 \mathbf{I}), \tag{9}$$

*where $\boldsymbol{\mu}_S = \mathbb{E}_{t \sim U(0,T)} \boldsymbol{\mu}_t$ with $\boldsymbol{\mu}_t = \frac{\epsilon \mathbf{1}}{\gamma_t^2 \sigma_\mathbf{x}^2 + \sigma_t^2}$ and $\sigma_S^2 = \mathbb{E}_{t \sim U(0,T)} \sigma_t^2$ with $\sigma_t^2 = \frac{1}{\gamma_t^2 \sigma_\mathbf{x}^2 + \sigma_t^2}$.*

Theorem 1 tells us: 1) the first two findings show that the mean of EPS for the adversarial sample differs from that of the natural sample due to the additional perturbation term $\boldsymbol{\mu}_S$; 2) the third finding indicates that the EPS of the natural sample is closer to that of other natural samples compared to adversarial samples, and this discrepancy becomes more pronounced when the perturbation transition $\gamma_t$ and $\sigma_t$ are small. These findings motivate us to employ $S(\mathbf{x})$ for adversarial detection.

**Why multiple scores?** Note that $\boldsymbol{\mu}_t$ and $\sigma_t^2$ decrease as the timestep $t$ increases due to the increase of $\gamma_t$ and $\sigma_t$. However, smaller variance $\sigma_S^2$ and larger mean $\mu_S$ are required for good adversarial detection. If we only consider one score of some unique timestep $t$ (i.e., removing the expectation from the definition of EPS), the variance $\sigma_S^2$ and mean $\boldsymbol{\mu}_S$ of the discrepancy will be so fluctuant that performing detecting adversarial samples will be very sensitive to the timestep $t$ (as validated in Section 4.4). To alleviate this issue, we consider taking expectation w.r.t. the timestep on multiple scores. In this way, the distribution of the discrepancy between the natural sample and the adversarial sample will be more stable to the timestep, which makes it easier to obtain a superior solution.

### 3.3 Adversarial Detection with Expected Perturbation Score

**Estimation for expected perturbation score.** Note that EPS (*i.e.*, $S(\mathbf{x})$) in Eq. (7) requires knowing the score function $\nabla_\mathbf{x} \log p_t(\mathbf{x})$, which can be estimated by training a neural network with score matching (Hyvärinen & Dayan, 2005; Song & Ermon, 2019; Vincent, 2011; Kingma et al., 2021). To this end, we model the perturbation process transition as a Gaussian distribution $p_{0t}(\mathbf{x}_t \mid \mathbf{x}_0) = \mathcal{N}(\mathbf{x}_t; \gamma_t \mathbf{x}_0, \sigma_t^2 \mathbf{I})$, where $\gamma_t = e^{-\frac{1}{2} \int_0^t \beta(s) ds}$ and $\sigma_t^2 = 1 - e^{-\int_0^t \beta(s) ds}$ with $\beta(t)$ for $t \in [0, 1000]$ being the time-dependent noise schedule. By optimizing Eq. (4), with sufficient data and model capacity, score matching ensures that the optimal solution to Eq.(4) equals $\nabla_\mathbf{x} \log p_t(\mathbf{x})$ for almost $\mathbf{x}$ and $t$ (Song et al., 2021). As a result, the score $\nabla_\mathbf{x} \log p_t(\mathbf{x})$ can be approximated by $s_\theta(\mathbf{x}_t, t)$. In practice, we use a pre-trained diffusion model to achieve the estimation for the score.

**MMD for characterizing expected perturbation score.** Using the norm of estimated EPS $S(x)$ as a characterization for adversarial detection is straightforward. Nevertheless, using the norm of $S(x)$ as the criterion can only describe the magnitude of the EPS vector, so it ignores the rich information that can be derived from its direction. It is critical that we design a distance metric to measure the distance between the EPS of an upcoming sample and the EPSs of natural samples in order to derive more useful information from $S(x)$.

Benefiting from the superior performance of maximum mean discrepancy (MMD) in measuring two given distributions (Long et al., 2015; Zhu et al., 2019; Gao et al., 2021), we resort to it for characterizing EPS. The basic idea of MMD is that two distributions would be identical if two random variables are identical for any order, and the moment that makes the largest distance between the two distributions should be the measure of the two distributions when the two distributions are

not the same (Smola et al., 2007; Gong et al., 2022). Formally, we define two distributions for the natural samples $\{\mathbf{x}^{(i)}\}_{i=1}^n$ as $\mathbb{P}_X = \frac{1}{n}\sum_{i=1}^n \delta(\|\mathbf{x} - \mathbf{x}^{(i)}\|)$ and a test sample $\tilde{\mathbf{x}}$ as $\mathbb{Q}_Y = \delta(\|\mathbf{y} - \tilde{\mathbf{x}}\|)$, respectively, where $\delta$ is the Dirac function (Alt, 2006). Then we estimate the distance between these two distributions as

$$\widehat{\mathrm{MMD}}_b^2[\mathbb{P}_X, \mathbb{Q}_Y; \mathcal{H}_k] = \left[\frac{1}{n^2}\sum_{i,j=1}^n k\left(\mathbf{x}^{(i)}, \mathbf{x}^{(j)}\right) - \frac{2}{n}\sum_{i=1}^n k\left(\mathbf{x}^{(i)}, \mathbf{y}\right) + k(\mathbf{y}, \mathbf{y})\right], \qquad (10)$$

where $k(\mathbf{x}, \mathbf{y}) = \kappa(S(\mathbf{x}), S(\mathbf{y}))$ with $\kappa$ being the kernel of a reproducing kernel Hilbert space $\mathcal{H}_k$ such as the Gaussian kernel $\kappa(\mathbf{x}, \mathbf{y}) = \exp\left(-\|\mathbf{x} - \mathbf{y}\|^2 / (2\sigma^2)\right)$. In our practice, we consider deep kernel MMD following Liu et al. (2020).

## 3.4 THEORETICAL ANALYSIS FOR EPS-AD

Note that after performing the same perturbation process, the first and the third terms in Eq. (10) are the same for each upcoming sample, thus we only focus on the cross-term $J = \frac{2}{n}\sum_{i=1}^n k\left(\mathbf{x}^{(i)}, \mathbf{y}\right)$. Next, we analyze this term for the adversarial sample.

**Corollary 1.** *Considering the Gaussian kernel $\kappa(\mathbf{x}, \mathbf{y}) = \exp\left(-\|\mathbf{x} - \mathbf{y}\|^2 / (2\sigma^2)\right)$ and the assumption in Theorem 1, for $\forall 0 < \eta < 1$, the probability of $P\{k(\mathbf{x}, \hat{\mathbf{y}}) > \eta\}$ is given by*

$$P\{k(\mathbf{x}, \hat{\mathbf{y}}) > \eta\} = \int_0^C \chi_d^2(\|\boldsymbol{\mu}_S\|^2)dx, \qquad (11)$$

*where $\boldsymbol{\mu}_S$ denotes the mean of $S(\mathbf{x}) - S(\hat{\mathbf{y}})$, $C$ is a constant for given $\eta$ and $\sigma$, $\chi^2$ is the probability density function of noncentral chi-squared distribution with $d$ degrees of freedom (Abdel-Aty, 1954).*

Corollary 1 indicates that the cross-term $J$ will be larger if $\|\boldsymbol{\mu}_S\|$ is close to zero given an $\eta$. Combining Eq. (8) and Eq. (9), we conclude that the natural data have larger $J$ than the adversarial data with higher probability due to the additional term $\mathbb{E}_t \frac{\epsilon \mathbf{1}}{\gamma_t^2 \sigma_{\mathbf{x}}^2 + \sigma_t^2}$, suggesting that the MMD between EPSs of the natural samples is smaller than that between natural and adversarial samples.

# 4 EXPERIMENTS

## 4.1 EXPERIMENTAL SETTINGS

**Datasets and network architectures.** We evaluate our method on CIFAR-10 (Krizhevsky, 2009) and ImageNet (Deng et al., 2009). We implement three widely used architectures as classifiers: WideResNet (Zagoruyko & Komodakis, 2016; Gowal et al., 2021) for CIFAR-10, ResNet (He et al., 2016) and ViT (Dosovitskiy et al., 2021) for ImageNet. For diffusion models, we consider the pre-trained diffusion models of Score SDE (Song et al., 2021) for CIFAR-10 and Guided Diffusion (Dhariwal & Nichol, 2021) for ImageNet, respectively.

**Attack methods.** Following Deng et al. (2021), we evaluate our adversarial detection method under various attack methods. We consider the commonly used $\ell_2$ and $\ell_\infty$ threat models, including PGD (Madry et al., 2018), FGSM (Goodfellow et al., 2015), BIM (Kurakin et al., 2018), MIM (Dong et al., 2018), TIM (Dong et al., 2019), CW (Carlini & Wagner, 2017), DI_MIM(Xie et al., 2019). Moreover, we apply two adaptive attack methods such as AutoAttack (AA) (Croce & Hein, 2020) and Minimum-Margin Attack (MM) (Gao et al., 2022). To show the superiority of our method, we consider the relatively low attack intensities, *i.e.*, $\ell_2$-ball and $\ell_\infty$-ball $\epsilon$-constrains with $\epsilon = 4/255$, and iterative attacks run for $5$ steps using step size $\epsilon/5$, unless stated otherwise.

**Baselines.** We compare our method with several state-of-the-art adversarial detection methods, including kernel density (KD) (Feinman et al., 2017), local intrinsic dimensionality (LID) (Ma et al., 2018), mahalanobis distance (MD) (Lee et al., 2018) and LiBRe (Deng et al., 2021). Besides, we construct two new adversarial detection methods based on diffusion models: 1) S-N: using the score norm of raw images, *i.e.*, $\|s_\theta(\mathbf{x}, t)\|^2$. 2) EPS-N: using the norm of the EPS, *i.e.*, $\|S(\mathbf{x})\|^2$. Differently, our proposed EPS-AD further calculates the *maximum mean discrepancy* of EPSs.

Table 1: Comparison of different adversarial detection methods on CIFAR-10 and ImageNet in terms of AUROC under $\epsilon = 4/255$. The bold number indicates the best results.

| Dataset | AUROC | FGSM | PGD | BIM | CW | FGSM-$\ell_2$ | BIM-$\ell_2$ | AA |
|---|---|---|---|---|---|---|---|---|
| CIFAR-10 | KD | 0.9213 | 0.9007 | 0.9037 | 0.8335 | 0.9121 | 0.9107 | 0.9135 |
| | LID | 0.9236 | 0.9159 | 0.9217 | 0.9014 | 0.9169 | 0.9320 | 0.9419 |
| | MD | 0.9990 | 0.9860 | 0.9758 | 0.9829 | **0.9995** | 0.9586 | 0.9830 |
| | S-N | **1.0000** | 0.9998 | 0.9961 | 0.9998 | 0.9885 | 0.9674 | 0.9995 |
| | EPS-N (Ours) | **1.0000** | **1.0000** | 0.9996 | **1.0000** | 0.9916 | 0.9883 | **1.0000** |
| | EPS-AD (Ours) | **1.0000** | **1.0000** | 0.9998 | **1.0000** | **0.9995** | 0.9991 | **1.0000** |
| ImageNet | KD | 0.7099 | 0.9720 | 0.9797 | 0.9413 | 0.7004 | 0.9775 | 0.9319 |
| | LID | 0.8912 | 0.9750 | 0.9808 | 0.9528 | 0.8932 | 0.9816 | 0.9582 |
| | MD | 0.8786 | 0.9773 | 0.9835 | 0.9609 | 0.8715 | 0.9830 | 0.9518 |
| | LiBRe | 0.9889 | 0.9530 | 0.9269 | 0.9039 | 0.8708 | 0.9211 | 0.8724 |
| | S-N | 0.9828 | 0.8974 | 0.7208 | 0.8969 | 0.8830 | 0.6762 | 0.9151 |
| | EPS-N (Ours) | 0.9987 | 0.9978 | 0.9215 | 0.9978 | 0.8191 | 0.7172 | 0.9943 |
| | EPS-AD (Ours) | **1.0000** | **1.0000** | **1.0000** | **1.0000** | **1.0000** | **1.0000** | **1.0000** |

(a) Attack Method: PGD  (b) Attack Method: FGSM  (c) Attack Method: CW

(d) Attack Method: BIM  (e) Attack Method: FGSM-$\ell_2$  (f) Attack Method: AA Attack

Figure 3: Comparison with adversarial detection methods on CIFAR-10 in terms of AUROC under $\epsilon \in \{1/255, \cdots, 8/255\}$. Sub-figures (a) - (f) share the same legend presented in sub-figure (a).

**Evaluation metric.** We evaluate the performance of adversarial detection approaches with the area under the receiver operating characteristic (AUROC), which is a widely used statistic for assessing the discriminatory capacity of distribution models (Jiménez-Valverde, 2012). Considering the computational cost of applying 12 attacks to the classifier, especially for the ImageNet, following Gao et al. (2021), we randomly select two disjoint subsets as adversarial and natural samples (each containing 500 samples) and compute the AUROC value over these two subsets. Notably, our method is applicable to different data set sizes, which is verified in Appendix. Moreover, we set $T^* = 20$ in $S(x)$ on CIFAR-10 and $T^* = 50$ on ImageNet for both our EPS-AD and EPS-N, and set $t^* = 5$ in $s_\theta(\mathbf{x}, t)$ on CIFAR-10 and $t^* = 20$ on ImageNet for S-N.

## 4.2 Detecting on Known Attacks

We start by comparing EPS-AD with the state-of-the-art adversarial detection methods that are trained with seen adversarial examples, against $\ell_2$ and $\ell_\infty$ threat models. Considering adversarial detection becomes more challenging when the attack intensity of adversarial samples is low, to

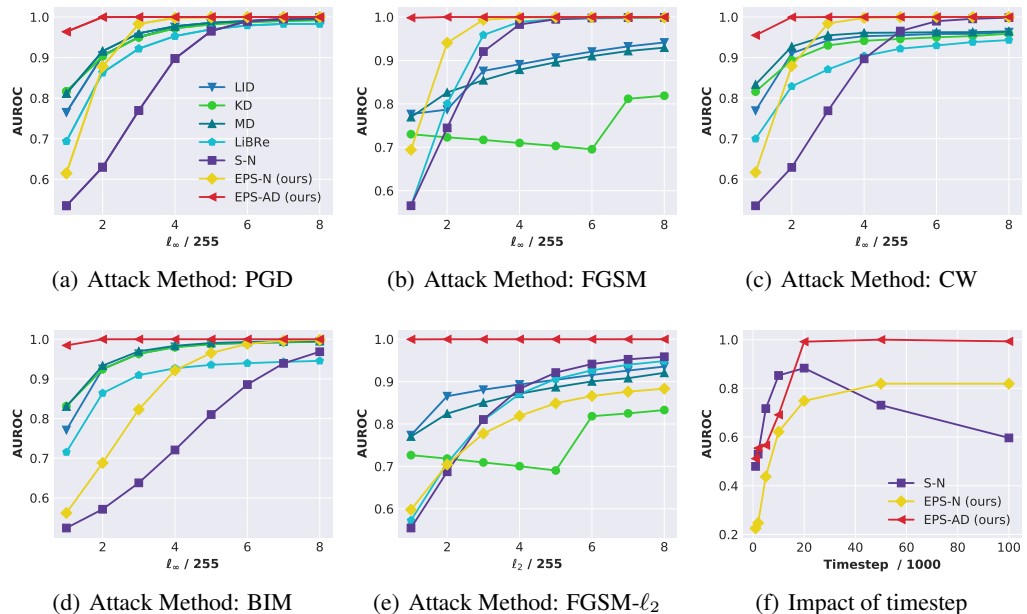

Figure 4: Results of adversarial detection on ImageNet. Sub-figures (a) - (e) report the AUROC on different attacks under $\epsilon \in \{1/255, \cdots, 8/255\}$ and share the same legend presented in sub-figure (a). Sub-figure (f) reports the AUROCs of different timesteps in $\{1, 2, 5, 10, 20, 50, 100\}$.

broadly evaluate the adversarial detection performance, we compare our method with other baselines on different attacks under different attack intensities. Moreover, to show the best performance of KD, LID and MD, we test the detection performance on their corresponding adversarial examples.

**Results on CIFAR-10.** Figure 3 shows our adversarial detection performance against 6 attacks under different attack intensities $\epsilon \in \{1/255, \cdots, 8/255\}$ on CIFAR-10 over WideResNet-28-10 (Zagoruyko & Komodakis, 2016) compared to other baselines. We demonstrate other attacks and more results on WideResNet-70-16 (Gowal et al., 2021) in Appendix. Obviously, our EPS-AD have much higher AUROC performance than other methods. Critically, we observe that our EPS-AD preserves almost non-degraded AUROC when the attack intensity $\epsilon$ surpasses $2/255$ against $\ell_2$ and $\ell_\infty$ attacks, which shows the stability of EPS-AD when detecting challenge adversarial samples.

In addition, we report quantitative results for adversarial detection under the attack intensity $\epsilon = 4/255$ in Table 1. The results show that by employing EPS and measuring their MMD, EPS-AD consistently outperforms existing methods against various attacks in terms of AUROC. We also see that by simply applying the norm of EPS, EPS-N achieves superior adversarial detection performance, which demonstrates the effectiveness of EPS.

**Results on ImageNet.** We report the adversarial detection performance against $\ell_2$ and $\ell_\infty$ attacks on ImageNet over ResNet-50 (He et al., 2016) in Table 1 and Figure 4 . We defer the results of one widely-used ViT architecture, DeiT-S (Dosovitskiy et al., 2021), in Appendix. We observe that our approach consistently outperforms baselines under various attacks, especially for detecting PGD, BIM and FGSM-$\ell_2$ attacks. These results reveal that our proposed EPS-AD is effective even on a large-scale data set. Moreover, we observe that EPS-N exhibits poor results compared to EPS-AD when detecting $\ell_2$ attacks (*e.g.*, FGSM-$\ell_2$ and BIM-$\ell_2$) since the norm of EPS ignores rich information contained in the EPS vector, which is not effective enough to detect on a large-scale data set. This performance degradation is more pronounced in the method S-N.

### 4.3 DETECTING ON UNSEEN ATTACKS AND TRANSFERABLE ATTACKS

In light of poor performance for adversarial detection baselines against unseen attacks and transferable attacks, we evaluate our method in the context of these kinds of attacks.

**Detecting on unseen attacks.** To detect unseen attacks, we train KD, LID, and MD detectors on CIFAR-10 using only FGSM and FGSM-$\ell_2$ adversarial examples and evaluate their performance

Table 2: Comparison of AUROC for detecting unseen attacks on CIFAR-10, where "FGSM (seen)" denotes the seen adversarial attack used for the training of KD, LID and MD.

| | FGSM(seen) | PGD | BIM | CW | FGSM-$\ell_2$ (seen) | BIM-$\ell_2$ | AA |
|---|---|---|---|---|---|---|---|
| KD | 0.9213 | 0.9007 | 0.9082 | 0.8339 | 0.9146 | 0.9146 | 0.9135 |
| LID | 0.9236 | 0.8964 | 0.9028 | 0.8828 | 0.9160 | 0.8984 | 0.9253 |
| MD | 0.9990 | 0.9855 | 0.9742 | 0.9835 | 0.9992 | 0.9503 | 0.9820 |
| S-N | **1.0000** | 0.9998 | 0.9961 | 0.9998 | 0.9885 | 0.9674 | 0.9995 |
| EPS-N (Ours) | **1.0000** | **1.0000** | 0.9996 | **1.0000** | 0.9916 | 0.9883 | **1.0000** |
| EPS-AD (Ours) | **1.0000** | **1.0000** | **0.9998** | **1.0000** | **0.9995** | **0.9991** | **1.0000** |

Table 3: Comparison of AUROC for detecting transferable attacks on ImageNet, where KD, LID, MD and LiBRe are trained with adversarial examples with ResNet-50 but detect the adversarial examples crafted with ResNet-101.

| | FGSM | PGD | BIM | CW | FGSM-$\ell_2$ | BIM-$\ell_2$ | AA |
|---|---|---|---|---|---|---|---|
| KD | 0.7754 | 0.5999 | 0.5847 | 0.7632 | 0.7906 | 0.7756 | 0.7698 |
| LID | 0.8467 | 0.7627 | 0.7663 | 0.7704 | 0.8520 | 0.7925 | 0.7967 |
| MD | 0.8467 | 0.7698 | 0.7684 | 0.7665 | 0.8067 | 0.7759 | 0.7880 |
| *Li*BRe | 0.9849 | 0.8414 | 0.7161 | 0.8286 | 0.8489 | 0.7250 | 0.8485 |
| S-N | 0.9816 | 0.8965 | 0.7166 | 0.8963 | 0.8764 | 0.6705 | 0.9106 |
| EPS-N (Ours) | 0.9983 | 0.9975 | 0.9178 | 0.9979 | 0.8235 | 0.7215 | 0.9930 |
| EPS-AD (Ours) | **1.0000** | **1.0000** | **1.0000** | **1.0000** | **1.0000** | **0.9998** | **1.0000** |

under other attacks. Combining Table 2 and 1, we find that adversarial detection performance for MD and LID worsens. An explanation is that their detectors are trained with vectorized features extracted from the seen samples through logistic regression, resulting in their limited generalization on unseen attacks. In contrast, diffusion-based detection methods show superior performance since they directly model the distribution of natural data.

**Detecting on transferable attacks.** To validate the transferability, we train KD, LID, MD and LiBRe detectors with ResNet-50 but detect the adversarial examples from a surrogate ResNet-101 model. Comparing Table 3 and 1, the non-diffusion-based methods (*e.g.*, KD, LID, MD and LiBRe) drop significantly against transferable attacks. By contrast, our EPS-AD method achieves significantly better transferability, since it does not rely on specific classifiers, but rather models the distribution of natural data, indicating its versatility in various attack scenarios.

### 4.4 ABLATION STUDY ON IMPACT OF TIMESTEP

We conduct experiments on ImageNet over ResNet-50 to show the effect of the timestep. To this end, we set the total timestep $T = 100$, which is sufficient for both EPS-AD and EPS-N to achieve a good solution. As shown in Figure 4 (f), we draw two observations: 1) our EPS-AD and EPS-N are insensitive to the total timestep $T$ while S-N fluctuates greatly with the timestep $t$; 2) As the total timestep $T$ increases, EPS-AD and EPS-N exhibit progressively better performance, however, this gain gradually decreases when $T$ exceeds the optimal value. This is due to the larger diffusion time, the mean $\mu_S$ in Eq.(9) gradually approaches zeros, resulting in a smaller discrepancy between the natural and adversarial distributions. Nie et al. (2022) also confirms this phenomenon (Theorem 1).

### 5 CONCLUSION

In this paper, we propose a new statistic *expected perturbation score* (EPS) to capture the information from multi-view observations of one sample, which is able to distinguish between natural and adversarial data well. Relying on EPS, we propose a novel adversarial detection method, EPS-AD. We provide theoretical analysis to demonstrate the superiority of EPS-AD. Extensive experiments on CIFAR-10 and ImageNet across different network architectures including ResNet, WideResNet and ViT show our EPS-AD successfully detects adversarial samples in various attack scenarios.

## REPRODUCIBILITY STATEMENT

To ensure the reproducibility of experimental results, we provide an exhaustive implementation in Section D and the code will be released upon acceptance.

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

# APPENDIX OF EPS-AD

In appendix, we provide detailed proofs of the theorem and corollary, descriptions of related works, more details and more experimental results of the proposed EPS-AD. We organize the supplementary into the following sections. In Section A, we provide descriptions of related works regarding adversarial detection. In Section B, we derive the proofs of the theorem and corollary. In Section C, we demonstrate the pseudo-code of our proposed EPS-AD. In Section D, we present detailed implementation of our experiments. In Section E, we show the impact of EPS in our method. In Section F, we study the impact of adding perturbations over samples in our method. In Section G, we report more comparison results, more ablation studies and more applications of our proposed EPS-AD.

## A  RELATED WORK

**Diffusion models.**   Diffusion models have emerged as a powerful generative model in many synthesis tasks (Song & Ermon, 2019; Dhariwal & Nichol, 2021; Saharia et al., 2022; Rombach et al., 2022; Ramesh et al., 2022). Since then, many researchers exploit the diffusion model to adversarial purification for improving the robustness of model (Nie et al., 2022; Yoon et al., 2021), where the score becomes a powerful means. Yet, only few researchers apply them to adversarial detection. Recently, Yoon et al. (2021) employ the score model to purify the adversarial examples and use the norm of scores as the stopping condition of the purification. They also demonstrate some results about the norm of score in distinguishing adversarial samples from natural samples. However, this criterion is not effective enough for adversarial detection. In our work, we comprehensively consider multiple scores of perturbed samples, where these perturbed samples are from the same sample, and exploit rich information from these scores for adversarial detection. We empirically compare our approach to these methods and find that our approach outperforms these methods by a large margin.

**Adversarial Attack.**   Numerous studies have been proposed to attack neural networks by slightly modifying the input data to trigger misclassifications of classifiers. We enumerate a series of such works in what follows. Fast Gradient Sign Method (FGSM) (Goodfellow et al., 2015) simply adds small noise along the gradient of the loss function. To further adjust the direction of increment, Kurakin et al. (2018) propose Basic Iterative Method (BIM) that extends FGSM to iteratively take multiple steps. Madry et al. (2018) propose Projected Gradient Descent (PGD) by combining the iterative method with random initialization for the adversarial example. Meanwhile, (Dong et al., 2018) propose Momentum Iterative Method (MIM) by adding a momentum term to BIM for achieving a more stable attack. Besides, Dong et al. (2019) propose Translation-Invariant Method (TIM) by optimizing the perturbation over translated images to obtain more transferable attacks, which can be incorporated into the methods FGSM and BIM. To further improve the transferability of attack, Wang & He (2021) propose variance tunning to guide the gradient update, namely VMI-FGSM; Zhang et al. (2022) conduct feature-level attacks with more accurate neuron importance estimations, called Neuron Attribution-based (NAA) attack. Besides the non-targeted attack methods mentioned above, there are also many methods that perturb data to one target label. For example, Carlini & Wagner (2017) perform an attack by incorporating the iterative mechanism of BIM, called CW. Besides, there are approaches that combine multiple attacks to perturb data, such as the AutoAttack (Croce & Hein, 2020) and the Minimum-margin (MM) attack (Gao et al., 2022), the faster version of AutoAttack. The various attacks may cause serious consequences in security-critical tasks, raising an urgent requirement for advanced techniques to achieve a robust model.

**Adversarial detection.**   To ensure the safety of machine learning system, a plethora of exploration for adversarial detection has attracted increasing sight. The most common idea is to filter out adversarial samples from test data using a trained binary classifier. Recently, statistics on hidden-layer features of DNNs are widely considered for adversarial detection. Feinman et al. (2017) train a logistic regression classifier using Kernel Density (KD) of features in the last hidden layer, as well as Bayesian Uncertainty (BU) as a basis. Ma et al. (2018) consider the local intrinsic dimensionality (LID) of the features of DNNs as the characteristics for detection. Lee et al. (2018) use a mahalanobis distance-based score to detect adversarial examples. In addition, Deng et al. (2021) train a Bayesian neural network by adding uniform noises to the samples. However, these methods train a tailored detector for some specific attacks or for a specific classifier, which largely overlooks the modeling for data distribution, leading to their limited performance under unknown attacks.

# B  PROOFS IN SECTION 3

## B.1  PROOF OF THEOREM 1

**Theorem 1** *Assuming that the distribution of natural data $p(\mathbf{x}) = \mathcal{N}(\boldsymbol{\mu}_{\mathbf{x}}, \sigma_{\mathbf{x}}^2\mathbf{I})$, given a perturbation transition kernel $p_{0t}(\mathbf{x}_t \mid \mathbf{x}_0) = \mathcal{N}(\mathbf{x}_t; \gamma_t\mathbf{x}_0, \sigma_t^2\mathbf{I})$ with $\gamma_t$ and $\sigma_t$ being the time-dependent noise schedule, then the following three conclusions hold for $S(\mathbf{x})$:*
*1) For $\forall\, \mathbf{x}{\sim}p(\mathbf{x})$, $S(\mathbf{x}) \sim \mathcal{N}(\mathbf{0}, \sigma_S^2\mathbf{I})$;*
*2) For $\forall\, \mathbf{y}{\sim}p(\mathbf{x})$, adversarial sample $\hat{\mathbf{y}}{=}\mathbf{y}{+}\epsilon$ with perturbation $\epsilon$, $S(\hat{\mathbf{y}}){\sim}\mathcal{N}(-\boldsymbol{\mu}_S, \sigma_S^2\mathbf{I})$;*
*3) For $\forall\, \mathbf{x}, \mathbf{y} \sim p(\mathbf{x})$, adversarial sample $\hat{\mathbf{y}}{=}\mathbf{y}{+}\epsilon$, we have*

$$S(\mathbf{x}) - S(\mathbf{y}) \xrightarrow{d} \mathcal{N}(\mathbf{0}, 2\sigma_S^2\mathbf{I}); \tag{12}$$

$$S(\mathbf{x}) - S(\hat{\mathbf{y}}) \xrightarrow{d} \mathcal{N}(\boldsymbol{\mu}_S, 2\sigma_S^2\mathbf{I}), \tag{13}$$

*where $\boldsymbol{\mu}_S{=}\mathbb{E}_t \frac{\epsilon\mathbf{1}}{\gamma_t^2\sigma_{\mathbf{x}}^2+\sigma_t^2}$ and $\sigma_S^2 = \mathbb{E}_t \frac{1}{\gamma_t^2\sigma_{\mathbf{x}}^2+\sigma_t^2}$.*

*Proof.* 1) Based on the distribution $p(\mathbf{x})$, *i.e.*, $\mathbf{x}_0 = \mathbf{x} \sim \mathcal{N}(\boldsymbol{\mu}_{\mathbf{x}}, \sigma_{\mathbf{x}}^2\mathbf{I})$, we obtain $\mathbf{x}_0 = \boldsymbol{\mu}_{\mathbf{x}} + \sigma_{\mathbf{x}}\mathbf{z}$ with $\mathbf{z} \sim \mathcal{N}(\mathbf{0}, \mathbf{I})$; based on the perturbation transition kernel $p_{0t}(\mathbf{x}_t \mid \mathbf{x}_0) = \mathcal{N}(\mathbf{x}_t; \gamma_t\mathbf{x}_0, \sigma_t^2\mathbf{I})$, we have $\mathbf{x}_t = \gamma_t\mathbf{x}_0 + \sigma_t\mathbf{z}$. Combining the distribution of $\mathbf{x}$ and $\mathbf{x}_0$, we have

$$\mathbf{x}_t = \gamma_t\boldsymbol{\mu}_{\mathbf{x}} + \sqrt{\gamma_t^2\sigma_{\mathbf{x}}^2 + \sigma_t^2}\mathbf{I}, \quad \textit{i.e.}, \mathbf{x}_t \sim \mathcal{N}(\gamma_t\boldsymbol{\mu}_{\mathbf{x}}, \gamma_t^2\sigma_{\mathbf{x}}^2 + \sigma_t^2) \tag{14}$$

For $\mathbf{x}_t \sim p_t(\mathbf{x}) = \mathcal{N}(\gamma_t\boldsymbol{\mu}_{\mathbf{x}}, \gamma_t^2\sigma_{\mathbf{x}}^2 + \sigma_t^2)$, we calculate the derivation

$$\nabla_{\mathbf{x}} \log p_t(\mathbf{x}) = -\frac{\mathbf{x}_t - \gamma_t\boldsymbol{\mu}_{\mathbf{x}}}{\gamma_t^2\sigma_{\mathbf{x}}^2 + \sigma_t^2} = -\frac{1}{\sqrt{\gamma_t^2\sigma_{\mathbf{x}}^2 + \sigma_t^2}}\mathcal{N}(\mathbf{0}, \mathbf{I}). \tag{15}$$

Taking expectation to $t$, we give the distribution of $S(\mathbf{x})$

$$S(\mathbf{x}) \sim \mathcal{N}(\mathbf{0}, \sigma_S^2\mathbf{I}), \quad \text{where } \sigma_S^2 = \mathbb{E}_t \frac{1}{\gamma_t^2\sigma_{\mathbf{x}}^2 + \sigma_t^2}. \tag{16}$$

2) Based on $\mathbf{y} \sim p(\mathbf{x})$ and $\hat{\mathbf{y}} = \mathbf{y} + \epsilon$, we obtain $\hat{\mathbf{y}}_0 = \hat{\mathbf{y}} \sim \mathcal{N}(\boldsymbol{\mu}_{\mathbf{x}} + \epsilon, \sigma_{\mathbf{x}}^2\mathbf{I})$. Then, we have

$$\nabla_{\hat{\mathbf{y}}} \log p_t(\hat{\mathbf{y}}) = -\frac{\mathbf{y}_t + \epsilon - \gamma_t\boldsymbol{\mu}_{\mathbf{x}}}{\gamma_t^2\sigma_{\mathbf{x}}^2 + \sigma_t^2} = -\frac{1}{\sqrt{\gamma_t^2\sigma_{\mathbf{x}}^2 + \sigma_t^2}}\mathcal{N}(\mathbf{0}, \mathbf{I}) - \frac{\epsilon}{\gamma_t^2\sigma_{\mathbf{x}}^2+\sigma_t^2}, \tag{17}$$

where the last equation is based on $p_t(\mathbf{y}) = p_t(\mathbf{x}) = \mathcal{N}(\gamma_t\boldsymbol{\mu}_{\mathbf{x}}, \gamma_t^2\sigma_{\mathbf{x}}^2 + \sigma_t^2)$.

Taking expectation to $t$, we give the distribution of $S(\hat{\mathbf{y}})$

$$S(\hat{\mathbf{y}}){\sim}\mathcal{N}(-\boldsymbol{\mu}_S, \sigma_S^2\mathbf{I}), \quad \text{where } \boldsymbol{\mu}_S{=}\mathbb{E}_t \frac{\epsilon\mathbf{1}}{\gamma_t^2\sigma_{\mathbf{x}}^2+\sigma_t^2} \text{ and } \sigma_S^2 = \mathbb{E}_t \frac{1}{\gamma_t^2\sigma_{\mathbf{x}}^2 + \sigma_t^2}. \tag{18}$$

3) According to the additive property of the Gaussian distribution, combing Eq. (16) and Eq. (18), we obtain the third conclusion.

$\square$

## B.2  PROOF OF COROLLARY 1

**Corollary 1** *Considering the Gaussian kernel $\kappa(\mathbf{x}, \mathbf{y}) = \exp\left(-\|\mathbf{x} - \mathbf{y}\|^2 / (2\sigma^2)\right)$ and the assumption in Theorem 1, for $\forall 0{<}\eta{<}1$, the probability of $P\{k(\mathbf{x}, \mathbf{y}) > \eta\}$ is given by*

$$P\{k(\mathbf{x}, \hat{\mathbf{y}}){>}\eta\} = \int_0^C \chi_d^2(\|\boldsymbol{\mu}_S\|^2)dx, \tag{19}$$

*where $\boldsymbol{\mu}_S$ denotes the mean of $S(\mathbf{x}) - S(\hat{\mathbf{y}})$, $C$ is a constant for given $\eta$ and $\sigma$, $\chi^2$ is the probability density function of noncentral chi-squared distribution with $d$ degrees of freedom (Abdel-Aty, 1954).*

*Proof.* Based on the Gaussian kernel $\kappa(\mathbf{x}, \mathbf{y})$, we have

$$P\{k(\mathbf{x}, \hat{\mathbf{y}}) > \eta\} = P\{\kappa(S(\mathbf{x}), S(\hat{\mathbf{y}})) > \eta\} = P\left\{\exp\left(-\|S(\mathbf{x}) - S(\hat{\mathbf{y}})\|^2 / (2\sigma^2)\right) > \eta\right\} \quad (20)$$

$$= P\left\{\|S(\mathbf{x}) - S(\hat{\mathbf{y}})\|^2 < -2\sigma^2 \ln \eta\right\}. \quad (21)$$

Let $\boldsymbol{\xi} \sim S(\mathbf{x}) - S(\hat{\mathbf{y}})$, then $\boldsymbol{\xi_i} \sim \mathcal{N}((\boldsymbol{\mu}_S)_i, \sigma_S^2)$, thus we have

$$P\{k(\mathbf{x}, \hat{\mathbf{y}}) > \eta\} = P\left\{\sum_{i=1}^d \boldsymbol{\xi_i}^2 < -2\sigma^2 \ln \eta\right\} = P\left\{\sum_{i=1}^d (\frac{\boldsymbol{\xi_i}}{\sigma_S})^2 < \frac{-2\sigma^2 \ln \eta}{\sigma_S^2}\right\} \quad (22)$$

Note that $\frac{\boldsymbol{\xi_i}}{\sigma_S} \sim \mathcal{N}((\boldsymbol{\mu}_S)_i, 1)$, based on the definition of noncentral chi-squared distribution (Abdel-Aty, 1954), we have

$$\sum_{i=1}^d (\frac{\boldsymbol{\xi_i}}{\sigma_S})^2 \sim \chi_d^2(\|\boldsymbol{\mu}_S\|^2) \quad (23)$$

Let $C = \frac{-2\sigma^2 \ln \eta}{\sigma_S^2}$, we obtain the conclusion

$$P\{k(\mathbf{x}, \hat{\mathbf{y}}) > \eta\} = \int_0^C \chi_d^2(\|\boldsymbol{\mu}_S\|^2) dx. \quad (24)$$

$\square$

## C PSEUDO-CODE OF EPS-AD

---

**Algorithm 1:** Expected Perturbation Score for Adversarial Detection (EPS-AD)

---

**Input:** Natural sample set $\{\mathbf{x}_0^{(i)}\}_{i=1}^n$, an upcoming sample $\tilde{\mathbf{x}}_0$, pre-trained score model $s_{\boldsymbol{\theta}}(\mathbf{x}_t, t)$ and diffusion timestep $T^*$.

**Output:** MMD between EPS of the upcoming sample $\tilde{\mathbf{x}}_0$ and EPSs of natural samples $\{\mathbf{x}_0^{(i)}\}$.

Set initial time step $t = 1$ ;

**for** $t = 1, \ldots, T^*$ **do**

 Obtaining perturbed natural samples $\{\mathbf{x}_t^{(i)}\}_{i=1}^n$ according to $p_{0t}(\mathbf{x}_t \mid \mathbf{x}_0)$;

 Obtaining perturbed upcoming sample $\tilde{\mathbf{x}}_t^{(i)}$ according to $p_{0t}(\hat{\mathbf{x}}_t \mid \tilde{\mathbf{x}}_0)$.

**end**

Compute EPSs of natural samples $\{S(\mathbf{x}^{(i)})\}_{i=1}^n$ using Eq. (7).

Compute EPS of the upcoming sample $S(\tilde{\mathbf{x}})$ using Eq. (7).

Compute the MMD between $\{S(\mathbf{x}^{(i)})\}_{i=1}^n$ and $S(\tilde{\mathbf{x}})$ using Eq. (10).

---

# D  MORE DETAILS FOR EXPERIMENT SETTINGS

## D.1  IMPLEMENTATION DETAILS OF OUR METHOD

Our adversarial detection method is built upon diffusion models. Specifically, we consider the pre-trained diffusion model of Score SDE for CIFAR-10 following Song et al. (2021) and choose the *vp/cifar10_ddpmpp_deep_continuous* checkpoint from the *score_sde* library [1]; for ImageNet, we consider the pre-trained diffusion model of Guided Diffusion following Dhariwal & Nichol (2021) and use the $256 \times 256$ *diffusion (unconditional)* checkpoint from the *guided-diffusion* library [2].

For classifiers, we use pre-trained WiderResNet-28-10 (Zagoruyko & Komodakis, 2016) and WiderResNet-70-16 (Gowal et al., 2021) for CIFAR-10, and ResNet-50, ResNet-101 (He et al., 2016), and DeiT-S (Dosovitskiy et al., 2021) for ImageNet. For attacks, we consider 8 attack intensities $\epsilon \in [1/255, 8/255]$ with iterative attacks run for 5 steps using step size $\epsilon/5$ under 12 different $\ell_2$ and $\ell_\infty$ attack methods to generate adversarial examples, including PGD, PGD-$\ell_2$ (Madry et al., 2018), FGSM, FGSM-$\ell_2$ (Goodfellow et al., 2015), BIM, BIM-$\ell_2$ (Kurakin et al., 2018), MIM (Dong et al., 2018), TIM (Dong et al., 2019), CW (Carlini & Wagner, 2017), DI_MIM(Xie et al., 2019) and two adaptive attacks AutoAttack (AA) (Croce & Hein, 2020) and Minimum-Margin Attack (MM) (Gao et al., 2022) that is a faster version of AA. And for evaluation, we choose the area under the receiver operating characteristic curve (AUROC) as metric for adversarial detection. Through all our experiments, we only use FGSM and FGSM-L2 ($\epsilon = 1/255$) to train a deep kernel to perform detection against all the attacks following Liu et al. (2020)[3], where the deep kernel can also be trained on a general public dataset which we leave for our future work. Note that our method is suitable for detecting all the $\ell_2$ and $\ell_\infty$ adversarial samples. We conduct our experiments based on Python 3.7 and Pytorch 1.7.1 on a server with 1× RTX 3090 GPU.

## D.2  IMPLEMENTATION DETAILS OF BASELINES

We choose three standard adversarial detection approaches, KD (Feinman et al., 2017), LID (Ma et al., 2018) and MD (Lee et al., 2018) as baselines for both CIFAR-10 and ImageNet, as well as LiBRe (Deng et al., 2021) individually on the ImageNet, which trains a Bayesian neural network by adding the uniform noises into the samples. Besides, we also construct two additional diffusion-based detection methods:1) S-N: using the score norm of raw images as the characteristics, *i.e.*, $||s_\theta(\mathbf{x}, t)||^2$; 2) EPS-N: using the norm of the EPS as the characteristics, *i.e.*, $||S(\mathbf{x})||^2$.

**KD & LID & MD.** We implement KD following the codebase[4], LID following the codebase[5], and MD following the codebasse[6]. These three methods train a logistic regressor to distinguish natural, noisy and adversarial samples. To show their best performance for adversarial detection, we choose the noise scale of the $\ell_2$ distance in KD, LID and MD as 40, 1, 10 on CIFAR-10 and 20, 1, 10 on ImageNet. Besides, for KD, we choose bandwidth as 10 on CIFAR-10 and 20 on ImageNet.

**LiBRe.** We implement LiBRe on ImageNet following their codebase [7]. We evaluate its performance on ResNet-50 and ResNet-101 to make a comparison.

## D.3  IMPLEMENTATION DETAILS OF FIGURE 1

For the results in Figure 1, we calculate the norms of scores of natural samples and adversarial samples at different purification timesteps using a score model pre-trained on ImageNet, where these adversarial samples are crafted by FGSM with $\epsilon = 1/255$. Before feeding these samples into the score model, we do not perturb them again. To better demonstrate these results, we normalize score norm with the maximum of scores in each timestep.

---

[1] https://github.com/yang-song/score_sde

[2] https://github.com/openai/guided-diffusion

[3] https://github.com/fengliu90/DK-for-TST

[4] https://github.com/rfeinman/detecting-adversarial-samples

[5] https://github.com/xingjunm/lid_adversarial_subspace_detection

[6] https://github.com/pokaxpoka/deep_Mahalanobis_detector

[7] https://github.com/thudzj/ScalableBDL

# E IMPACT OF EPS FOR EPS-AD

In our method EPS-AD, we use diffusion-based score model to calculate the characteristics of samples, *i.e.*, EPS, which has the same dimension as this sample. In this experiment, we investigate the impact of EPS in our method. As a result, we remove the calculation of EPS from our method, instead using the raw sample as their characteristic. Table 4 shows adversarial detection performance of our method against 6 attacks under attack intensities $\epsilon \in \{2/255, 4/255\}$ on ImageNet over ResNet-50 compared to that without EPS. Obviously, EPS-AD without employing EPS demonstrates significant performance drop ($\approx 28\% \downarrow$), suggesting the superiority of our proposed EPS in distinguishing between adversarial and natural samples.

Table 4: Impact of EPS with ResNet-50 on ImageNet under $\epsilon = 2/255$ and $\epsilon = 4/255$.

| Perturbation | Method | FGSM | PGD | BIM | MIM | TIM | CW |
|---|---|---|---|---|---|---|---|
| $\epsilon = 2/255$ | EPS-AD (w/o EPS) | 0.7132 | 0.7108 | 0.7101 | 0.7119 | 0.7120 | 0.7107 |
| | EPS-AD (w/ EPS) | **1.0000** | **0.9997** | **0.9999** | **1.0000** | **0.9983** | **0.9995** |
| $\epsilon = 4/255$ | EPS-AD (w/o EPS) | 0.7176 | 0.7144 | 0.7131 | 0.7156 | 0.7156 | 0.7142 |
| | EPS-AD (w/ EPS) | **1.0000** | **1.0000** | **1.0000** | **1.0000** | **1.0000** | **1.0000** |
| Perturbation | Method | DI_MIM | PGD-$\ell_2$ | FGSM-$\ell_2$ | BIM-$\ell_2$ | MM | AA |
| $\epsilon = 2/255$ | EPS-AD (w/o EPS) | 0.7117 | 0.7118 | 0.7133 | 0.7097 | 0.7066 | 0.7067 |
| | EPS-AD (w/ EPS) | **0.9999** | **1.0000** | **1.0000** | **0.9999** | **1.0000** | **1.0000** |
| $\epsilon = 4/255$ | EPS-AD (w/o EPS) | 0.7156 | 0.7154 | 0.7175 | 0.7128 | 0.7101 | 0.7104 |
| | EPS-AD (w/ EPS) | **1.0000** | **1.0000** | **1.0000** | **1.0000** | **1.0000** | **1.0000** |

# F IMPACT OF ADDING PERTURBATIONS OVER SAMPLES FOR EPS-AD

Adding perturbations into the samples is critical for our proposed EPS-AD. To investigate the impact of this operation, we conduct ablation studies against 6 adversarial attack methods on ImageNet. Table 5 demonstrates adversarial detection performance of EPS-AD against 6 attacks under $\epsilon = 1/255$ on ImageNet over ResNet-50 compared to that without adding perturbations. We observe that the adversarial detection performance is significantly improved with adding perturbations. Specifically, our method obtains about $1.86\% \uparrow$ on average of 12 attacks, in which the maximum is $4.84\% \uparrow$ against BIM-$\ell_2$. This coincides with the conclusion in Theorem 1 that adding the perturbations helps distinguish between adversarial and natural samples.

Table 5: Impact of adding perturbations with ResNet-50 on ImageNet under $\epsilon = 1/255$.

| Method | PGD | BIM | CW | FGSM-$\ell_2$ | BIM-$\ell_2$ | AA |
|---|---|---|---|---|---|---|
| EPS-AD (w/o perturbation) | 0.9274 | 0.9405 | 0.9257 | 0.9960 | 0.9443 | 0.9835 |
| EPS-AD (w/ perturbation) | **0.9637** | **0.9845** | **0.9549** | **0.9997** | **0.9927** | **0.9936** |
| Method | FGSM | MIM | TIM | DI_MIM | PGD-$\ell_2$ | MM |
| EPS-AD (w/o perturbation) | 0.9954 | 0.9885 | 0.9397 | 0.9791 | 0.9858 | 0.9818 |
| EPS-AD (w/ perturbation) | **0.9982** | **0.9972** | **0.9561** | **0.9817** | **0.9961** | **0.9928** |

## G   MORE RESULTS OF ADVERSARIAL DETECTION

To further evaluate the effectiveness of our proposed EPS-AD, in subsection G.1, we conduct more comparison experiments on detecting more adversarial attacks on ImageNet and CIFAR-10 datasets. To demonstrate the generalization of EPS-AD, we conduct more experiments on detecting unseen attacks and transferable attacks. In subsection G.2, we additionally study the mechanism of EPS-AD by reporting the impact of time steps, set size, low attack intensity, transferability across datasets and a real application detecting face anti-spoofing samples. We provide extra results of all compared experiments in Table 6, 7, 8 and Figure 5, 6, 7, as supplements of additional attack methods corresponding to the main body.

### G.1   MORE COMPARISON EXPERIMENTS

**More comparison results on basic setup.**   In Table 6, we provide additional attack results including MIM, TIM, DI_MIM, PGD-$\ell_2$, MM on CIFAR-10 and ImageNet. We observe that EPS-AD keeps dominant position under these attacks, EPS-N and S-N exhibit poor performance on ImageNet due to the fact that these two methods use the norm of vectors and thus overlook the rich information that can be deviated from the vector. Moreover, we find that in Figure 5 and Figure 6 , most adversarial detection methods suffer from the extremely low attack intensity (*e.g.*, $\epsilon = 1/255$). In contrast, our method EPS-AD still has promising detection performance (refer to Table 14 for more quantitative results).

Table 6: More results of different adversarial detection methods on CIFAR-10 and ImageNet in terms of AUROC under $\epsilon = 4/255$ (MIM, TIM, DI, MIM, PGD-$\ell_2$, MM).

| Dataset | Method | MIM | TIM | DI_MIM | PGD-$\ell_2$ | MM |
|---------|--------|------|------|--------|--------|------|
| CIFAR-10 | KD | 0.8863 | 0.8856 | 0.8459 | 0.9296 | 0.9116 |
| | LID | 0.9135 | 0.8781 | 0.8638 | 0.9426 | 0.9380 |
| | MD | 0.9890 | **0.9998** | 0.9846 | 0.9958 | 0.9836 |
| | S-N | 0.9993 | 0.9985 | 0.9980 | **1.0000** | 0.9994 |
| | EPS-N (Ours) | 0.9999 | 0.9996 | 0.9996 | **1.0000** | **0.9996** |
| | EPS-AD (Ours) | **1.0000** | 0.9993 | **0.9999** | **1.0000** | 0.9995 |
| ImageNet | KD | 0.9669 | 0.9317 | 0.7402 | 0.9809 | 0.9377 |
| | LID | 0.9665 | 0.9397 | 0.7658 | 0.9833 | 0.9575 |
| | MD | 0.9664 | 0.9365 | 0.8612 | 0.9858 | 0.9585 |
| | LiBRe | 0.9483 | 0.8772 | 0.9968 | 0.9532 | 0.8697 |
| | S-N | 0.9087 | 0.8310 | 0.9143 | 0.9505 | 0.9090 |
| | EPS-N (Ours) | 0.9778 | 0.9114 | 0.9779 | 0.9989 | 0.9939 |
| | EPS-AD (Ours) | **1.0000** | **1.0000** | **1.0000** | **1.0000** | **1.0000** |

**More comparison results on unseen and transferable attacks.**   We also compare our EPS-AD with KD, LID MD and LiBRe under 6 additional unseen or transferable attacks (MIM, TIM, DI_MIM, PGD-$\ell_2$, MM, VMI-FGSM (Wang & He, 2021)) to further evaluate the effectiveness of our method. In Table 7 and Table 8, Our approach consistently exhibits superior generalization compared to other baselines.

**More comparison results on CIFAR-10 over robust WideResNet-70-16 .**   We further compare our method with baselines on an adversarial trained classifier on CIFAR-10, *e.g.*, WideResNet-70-16 (Gowal et al., 2021) against various attacks. In Table 9, 10, We observe that diffusion-based detection methods are much better than other baselines trained with specific adversarial samples. One possible reason is that adversarial samples are difficult to deceive robust classifiers, which means that such adversarial samples are ineffective for training effective detectors.

**More comparison results on ImageNet over DeiT-S.**   We further make an attempt on Vision-transformer-based architecture (*i.e.* DeiT-S (Dosovitskiy et al., 2021)) on ImageNet. Considering

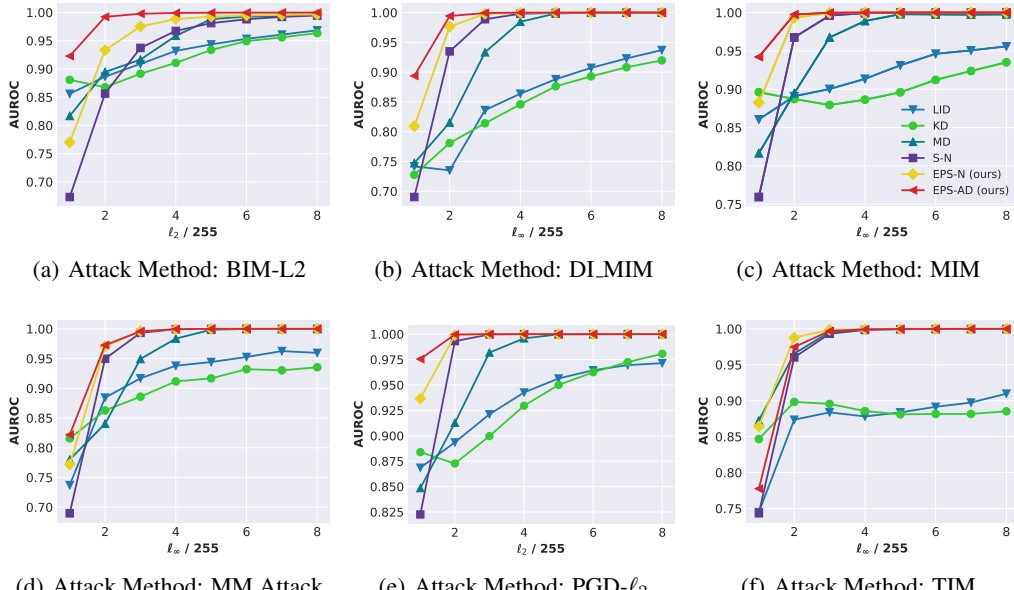

Figure 5: More Results of adversarial detection on CIFAR-10. Sub-figures (a) - (f) report the AUROC on different attacks under $\epsilon\in\{1/255,\cdots,8/255\}$ and share the same legend in sub-figure (c).

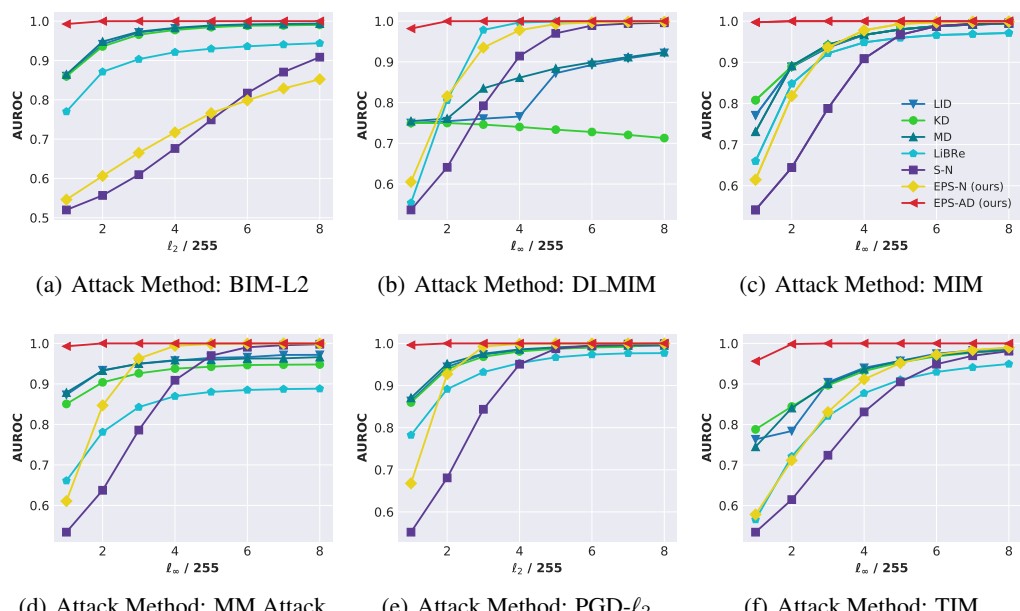

Figure 6: More Results of adversarial detection on ImageNet. Sub-figures (a) - (f) report the AUROC on different attacks under $\epsilon\in\{1/255,\cdots,8/255\}$ and share the same legend in sub-figure (c).

the specificity of the vision-transformer-based structure, we compare our method to three baselines (LID, S-N and EPS-N). From Table 11, our method exhibits consistent superiority when compared to other baselines, suggesting the versatility of EPS-AD with different architectures.

Table 7: More results of AUROC for detecting the unseen attacks (MIM, TIM, DI_MIM, PGD-$\ell_2$, MM) on CIFAR-10. "FGSM (seen)" denotes the seen adversarial attack used for the training of KD, LID and MD.

| Method | FGSM(seen) | MIM | TIM | DI_MIM | PGD-$\ell_2$ | MM |
|---|---|---|---|---|---|---|
| KD | 0.9213 | 0.8867 | 0.8876 | 0.8549 | 0.9303 | 0.9114 |
| LID | 0.9236 | 0.9131 | 0.8479 | 0.8631 | 0.9090 | 0.9244 |
| MD | 0.9990 | 0.9858 | 0.9998 | 0.9791 | 0.9958 | 0.9829 |
| S-N | **1.0000** | 0.9993 | 0.9985 | 0.9980 | **1.0000** | 0.9994 |
| EPS-N (Ours) | **1.0000** | 0.9999 | 0.9996 | 0.9996 | **1.0000** | 0.9996 |
| EPS-AD (Ours) | **1.0000** | **1.0000** | **0.9993** | **0.9999** | **1.0000** | **0.9995** |

Table 8: More results of AUROC for detecting the transferable attacks (MIM, TIM, DI_MIM, PGD-$\ell_2$, MM, VMI-FGSM) on ImageNet, where KD, LID, MD and LiBRe are trained with adversarial examples with ResNet-50 but detect the adversarial examples crafted with ResNet-101.

| Method | MIM | TIM | DI_MIM | PGD-$\ell_2$ | MM | VMI-FGSM |
|---|---|---|---|---|---|---|
| KD | 0.6355 | 0.7006 | 0.7558 | 0.7028 | 0.7381 | 0.7669 |
| LID | 0.7780 | 0.7978 | 0.7654 | 0.7734 | 0.7913 | 0.7386 |
| MD | 0.7756 | 0.7612 | 0.8395 | 0.7827 | 0.7864 | 0.8039 |
| *Li*BRe | 0.8966 | 0.7317 | 0.9722 | 0.8749 | 0.8388 | 0.9944 |
| S-N | 0.9049 | 0.8216 | 0.9117 | 0.9490 | 0.9043 | 0.8638 |
| EPS-N (Ours) | 0.9771 | 0.9049 | 0.9765 | 0.9987 | 0.9930 | 0.9609 |
| EPS-AD (Ours) | **1.0000** | **1.0000** | **1.0000** | **1.0000** | **1.0000** | **1.0000** |

## G.2 MORE DISCUSSIONS OF EPS-AD

**Ablation of timestep.** The ablation study of timestep is provided in Table 12. We observe that when the timestep $T$ stays in $[10, 100]$, EPS-AD and EPS-N both obtain usable AUROC in detecting FGSM-$\ell_2$ attack, which verifies that our method is insensitive to the timestep. Moreover, Our approach achieves optimum performance when $T^* = 20$ on CIFAR-10 and $T^* = 50$ on ImageNet.

**Impact of set size.** Previous adversarial detection methods usually measure the discrepancy well only with large amount of data (Gao et al., 2021). To show the effectiveness of our proposed EPS-AD, in this experiment, we further ablate the effect of set size by conducting experiments on 100 to 500 samples subset of CIFAR-10 with WideResNet-28-10 and ImageNet with ResNet-50. Performance of our three methods, S-N, EPS-N and EPS-AD, is shown in Table 13 and Figure 7. It is obvious that EPS-AD consistently outperforms EPS-N and S-N with small set size and large set size. Moreover, EPS-AD is robust to the changes of set size while EPS-N and S-N fluctuate with set size, especially on CIFAR-10 dataset.

**Computational efficiency of EPS-AD** Given a fixed-size model, the computational cost of our method (EPS-AD) mainly depends on two factors: the resolution of input images and total diffusion timestep $T$. Actually, our EPS-AD performs adversarial detection efficiently, especially on low-resolution images, and yields promising performance compared to existing methods. To evaluate the efficiency of EPS-AD, we randomly choose 500 images from CIFAR-10 and ImageNet respectively in detecting FGSM-$\ell_2$ adversarial samples on a single RTX3090 GPU. The average time costs per image for CIFAR-10 ($T = 20$) and ImageNet ($T = 50$) are $0.038s$ and $2.386s$, respectively.

To further demonstrate the effect of total diffusion timesteps on the efficiency, we provide the results under different diffusion timesteps against FGSM-L2 attack, as shown in Table 12, From the table, our EPS-AD method shows superior adversarial detection performance when $20 \leq T \leq 100$ on ImageNet and takes 0.954s when T = 20, which is much more efficient than that with T = 50. Moreover, our EPS-AD achieves superior or comparable performance on ImageNet and CIFAR-10 compared with existing methods (i.e., KD, LID, and MD) even with T as small as 20.

Table 9: Comparison of AUROC for using adversarial trained WideResNet-70-16 as classifier on CIFAR-10 under $\epsilon = 2/255$. Due to the constraint of memory and resources, we omit the detection results on AutoAttack for KD, LID and MD.

| Method | KD | LID | MD | S-N | EPS-N (Ours) | EPS-AD (Ours) |
|---|---|---|---|---|---|---|
| FGSM | 0.5852 | 0.7551 | 0.6924 | 0.9797 | 0.9976 | 0.9978 |
| PGD | 0.5672 | 0.7517 | 0.6846 | 0.9625 | 0.9954 | 0.9961 |
| BIM | 0.5786 | 0.7543 | 0.6787 | 0.9568 | 0.9935 | 0.9930 |
| MIM | 0.5795 | 0.7544 | 0.6804 | 0.9679 | 0.9963 | 0.9957 |
| TIM | 0.5812 | 0.7533 | 0.6783 | 0.8468 | 0.9506 | 0.9407 |
| CW | 0.5559 | 0.7511 | 0.6830 | 0.9600 | 0.9949 | 0.9953 |
| DI_MIM | 0.5763 | 0.7505 | 0.6712 | 0.8877 | 0.9728 | 0.9700 |
| PGD-$\ell_2$ | 0.6116 | 0.7632 | 0.7049 | 0.9942 | 0.9994 | 0.9998 |
| FGSM-$\ell_2$ | 0.6114 | 0.7619 | 0.7032 | 0.9471 | 0.9766 | 0.9861 |
| BIM-$\ell_2$ | 0.7550 | 0.7629 | 0.7060 | 0.9396 | 0.9756 | 0.9828 |

Table 10: Comparison of AUROC for using adversarial trained WideResNet-70-16 as classifier on CIFAR-10 under $\epsilon = 4/255$.

| Method | KD | LID | MD | S-N | EPS-N (Ours) | EPS-AD (Ours) |
|---|---|---|---|---|---|---|
| FGSM | 0.6020 | 0.7628 | 0.7668 | 0.9999 | **1.0000** | **1.0000** |
| PGD | 0.5913 | 0.7598 | 0.7535 | 0.9998 | **1.0000** | **1.0000** |
| BIM | 0.6076 | 0.7617 | 0.7588 | 0.9994 | **1.0000** | **1.0000** |
| MIM | 0.7683 | 0.7625 | 0.7601 | 0.9997 | **1.0000** | **1.0000** |
| TIM | 0.6029 | 0.7605 | 0.7563 | 0.9917 | 0.9984 | **0.9987** |
| CW | 0.5919 | 0.7581 | 0.7524 | 0.9998 | **1.0000** | **1.0000** |
| DI_MIM | 0.6015 | 0.7562 | 0.7551 | 0.9983 | 0.9998 | **0.9999** |
| PGD-$\ell_2$ | 0.8314 | 0.7774 | 0.7544 | **1.0000** | **1.0000** | **1.0000** |
| FGSM-$\ell_2$ | 0.7805 | 0.7709 | 0.7489 | 0.9803 | 0.9902 | **0.9967** |
| BIM-$\ell_2$ | 0.8351 | 0.7739 | 0.8141 | 0.9835 | 0.9927 | **0.9955** |

Note that the computational efficiency of our method can be further improved by applying an efficient sampling strategy (Lu et al., 2022), a low-resolution diffusion model (Dhariwal & Nichol, 2021) and a sparse diffusion timestep (e.g., sampling with a time interval of 2/1000 during the diffusion process). We leave these techniques for our future work.

**Detecting on low attack intensity.** To further reveal the superiority of our EPS-AD, we conduct an experiment under an extremely low attack intensity (*e.g.*, $\epsilon = 1/255$) on ImageNet. In Table 14, we observe that Our EPS-AD achieves a significant advantage in detecting adversarial samples crafted with extremely low attack intensity, demonstrating its significant effectiveness.

**Detecting on adversarial samples across datasets.** We further exploit the transferability across different datasets. To this end, we utilize a pre-trained score-based diffusion model on ImageNet to perform detecting adversarial samples from CIFAR-10. Specifically, we randomly select two disjoint subsets as adversarial and natural samples (each containing 500 samples) from CIFAR-10 and use a score model pre-trained on ImageNet to calculate the AUROC, which is named EPS-AD*. Table 15 demonstrates detection performance of 6 methods against 12 attacks under $\epsilon = 2/255$ on CIFAR-10 over WideResNet-28-10. We observe that EPS-AD* still exhibits superior detection performance compared to KD, LID, MD baselines, and achieves a comparable performance compared to other diffusion-based methods that use the score model pre-trained on CIFAR-10.

Table 11: Comparison of AUROC for using DeiT-S as classifier on ImageNet under $\epsilon = 4/255$.

| Method | LID | S-N | EPS-N (Ours) | EPS-AD (Ours) |
|---|---|---|---|---|
| FGSM | 0.8846 | 0.9789 | 0.9984 | **1.0000** |
| PGD | 0.9162 | 0.8935 | 0.9969 | **1.0000** |
| BIM | 0.9191 | 0.7331 | 0.9215 | **1.0000** |
| MIM | 0.9102 | 0.9025 | 0.9780 | **1.0000** |
| TIM | 0.9019 | 0.8091 | 0.8765 | **0.9606** |
| CW | 0.8742 | 0.8934 | 0.9975 | **0.9999** |
| DI_MIM | 0.7246 | 0.9074 | 0.9752 | **1.0000** |
| PGD-$\ell_2$ | 0.9041 | 0.9451 | 0.9987 | **1.0000** |
| FGSM-$\ell_2$ | 0.8698 | 0.7665 | 0.7023 | **1.0000** |
| BIM-$\ell_2$ | 0.9002 | 0.6564 | 0.6544 | **1.0000** |
| MM | 0.9164 | 0.8902 | 0.9886 | **0.9993** |
| AA | 0.9191 | 0.9023 | 0.9915 | **1.0000** |

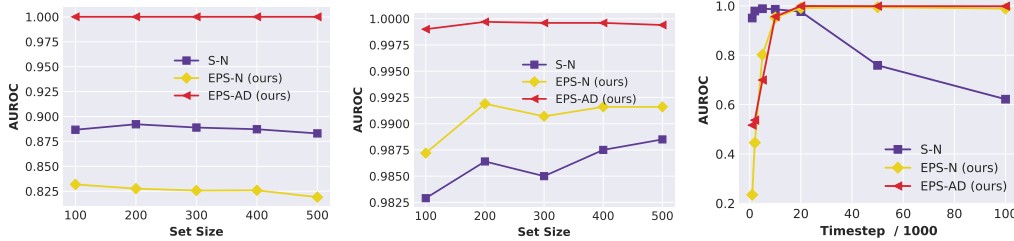

(a) Impact of set size on ImageNet    (b) Impact of set size on CIFAR-10    (c) Impact of timestep

Figure 7: Impact of different set sizes and diffusion time step. Sub-figures (a) and (b) report the AUROC on FGSM-$\ell_2$ attack under $\epsilon = 4/255$. Sub-figure (c) reports the AUROCs of different diffusion time in $\{1, 2, 5, 10, 20, 50, 100\}$ on CIFAR-10 dataset.

Table 12: Impact of timestep with WideResNet-28-10 on CIFAR-10 and ResNet-50 on ImageNet against FGSM-$\ell_2$ under $\epsilon = 4/255$.

| Dataset | timestep / Method | 1 | 2 | 5 | 10 | 20 | 50 | 100 |
|---|---|---|---|---|---|---|---|---|
| CIFAR-10 | S-N | 0.9508 | 0.9792 | 0.9885 | 0.9860 | 0.9768 | 0.7590 | 0.6218 |
| | EPS-N (Ours) | 0.2346 | 0.4454 | 0.8018 | 0.9526 | 0.9916 | 0.9915 | 0.9889 |
| | EPS-AD (Ours) | 0.5166 | 0.5368 | 0.6994 | 0.9566 | **0.9994** | 0.9988 | 0.9985 |
| ImageNet | S-N | 0.4794 | 0.5298 | 0.7168 | 0.8528 | 0.8830 | 0.7309 | 0.5964 |
| | EPS-N (Ours) | 0.2246 | 0.2466 | 0.4369 | 0.6215 | 0.7484 | 0.8191 | 0.8190 |
| | EPS-AD (Ours) | 0.5112 | 0.5545 | 0.5662 | 0.6912 | 0.9917 | **1.0000** | 0.9930 |

Table 13: Impact of data set size with WideResNet-28-10 on CIFAR-10 and ResNet-50 on ImageNet against FGSM-$\ell_2$ under $\epsilon = 4/255$.

| Dataset | size / Method | 100 | 200 | 300 | 400 | 500 |
|---|---|---|---|---|---|---|
| CIFAR-10 | S-N | 0.9829 | 0.9864 | 0.9850 | 0.9875 | 0.9885 |
| | EPS-N (Ours) | 0.9872 | 0.9919 | 0.9907 | 0.9916 | 0.9916 |
| | EPS-AD (Ours) | 0.9990 | 0.9997 | 0.9996 | 0.9996 | 0.9994 |
| ImageNet | S-N | 0.8867 | 0.8922 | 0.8889 | 0.8872 | 0.8830 |
| | EPS-N (Ours) | 0.8318 | 0.8276 | 0.8257 | 0.8259 | 0.8191 |
| | EPS-AD (Ours) | 1.0000 | 1.0000 | 1.0000 | 1.0000 | 1.0000 |

Table 14: Comparision of different adversarial detection methods with attack intensity $\epsilon = 1/255$ over ResNet-50 on ImageNet.

| Method | KD | LID | MD | LiBRe | S-N | EPS-N (Ours) | EPS-AD (Ours) |
|--------|------|------|------|-------|------|--------------|---------------|
| FGSM | 0.7301 | 0.7765 | 0.7694 | 0.5653 | 0.5654 | 0.6942 | **0.9982** |
| PGD | 0.8167 | 0.7651 | 0.8116 | 0.6938 | 0.5348 | 0.6149 | **0.9637** |
| BIM | 0.8313 | 0.7711 | 0.8303 | 0.7154 | 0.5240 | 0.5619 | **0.9845** |
| MIM | 0.8079 | 0.7711 | 0.7313 | 0.6597 | 0.5411 | 0.6144 | **0.9972** |
| TIM | 0.7879 | 0.7634 | 0.7449 | 0.5655 | 0.5344 | 0.5780 | **0.9561** |
| CW | 0.8161 | 0.7690 | 0.8336 | 0.6996 | 0.5345 | 0.6169 | **0.9549** |
| DI_MIM | 0.7498 | 0.7498 | 0.7543 | 0.5538 | 0.5364 | 0.6052 | **0.9817** |
| PGD-$\ell_2$ | 0.8597 | 0.8633 | 0.8707 | 0.7824 | 0.5519 | 0.6678 | **0.9961** |
| FGSM-$\ell_2$ | 0.7265 | 0.7737 | 0.7704 | 0.5727 | 0.5546 | 0.5978 | **0.9997** |
| BIM-$\ell_2$ | 0.8596 | 0.8602 | 0.8644 | 0.7701 | 0.5199 | 0.5462 | **0.9927** |
| MM | 0.8505 | 0.8730 | 0.8786 | 0.6611 | 0.5340 | 0.6106 | **0.9928** |
| AA | 0.8535 | 0.8706 | 0.8733 | 0.6629 | 0.5351 | 0.6113 | **0.9936** |

Table 15: Comparison of cross-dataset EPS-AD* under attack intensity $\epsilon = 2/255$ with other methods over WideResNet-28-10 on CIFAR-10, where EPS-AD* utilizes a score model pre-trained on ImageNet.

| Method | KD | LID | MD | S-N | EPS-N (Ours) | EPS-AD (Ours) | EPS-AD* |
|--------|------|------|------|------|--------------|---------------|---------|
| PGD | 0.8871 | 0.8836 | 0.8815 | 0.9679 | 0.9950 | 0.9972 | 0.9987 |
| FGSM | 0.9112 | 0.9010 | 0.9200 | 0.9902 | 0.9987 | 0.9994 | 0.9961 |
| BIM | 0.8786 | 0.8878 | 0.8811 | 0.9268 | 0.9811 | 0.9914 | 0.9890 |
| MIM | 0.8873 | 0.8909 | 0.8947 | 0.9676 | 0.9935 | 0.9975 | 0.9964 |
| TIM | 0.8983 | 0.8735 | 0.9655 | 0.9603 | 0.9878 | 0.9747 | 0.9906 |
| CW | 0.8634 | 0.8762 | 0.8968 | 0.9682 | 0.9953 | 0.9975 | 0.9971 |
| DI_MIM | 0.7810 | 0.7351 | 0.8153 | 0.9348 | 0.9764 | 0.9942 | 0.9918 |
| PGD-$\ell_2$ | 0.8727 | 0.8935 | 0.9128 | 0.9931 | 0.9989 | 0.9997 | 0.9994 |
| FGSM-$\ell_2$ | 0.9044 | 0.8927 | 0.9211 | 0.9166 | 0.9634 | 0.9976 | 0.9973 |
| BIM-$\ell_2$ | 0.8675 | 0.8864 | 0.8946 | 0.8564 | 0.9333 | 0.9922 | 0.9896 |
| MM | 0.8627 | 0.8843 | 0.8404 | 0.9497 | 0.9706 | 0.9727 | 0.9658 |
| AA | 0.8754 | 0.8894 | 0.8392 | 0.9595 | 0.9775 | 0.9819 | 0.9782 |

