# OpenReview forum: "Expected Perturbation Scores for Adversarial Detection"
_ICLR.cc/2023/Conference — Submitted to ICLR 2023_

### Official Review · Reviewer_yyTy · 2022-10-22

**Confidence:** 4
**Correctness:** 3
**Technical Novelty And Significance:** 3
**Empirical Novelty And Significance:** 3
**Recommendation:** 8

**Clarity, Quality, Novelty And Reproducibility:**

This paper introduces a natural and novel adversarial detection algorithm, which could benefit the community of adversarial attack and robustness. The experiments are sufficient. In terms of reproducibility, the description of the experiment setup is clear. More details are included in the appendix.

**Strength And Weaknesses:**

Pros:

++ The paper is well-written and well-motivated. Motivated by the ineffectiveness of the score of one sample, the authors propose to use expected perturbation score via pre-trained diffusion model seems natural.

++ The authors provide theoretical analysis to show that proposed expected perturbation score can better distinguish the distributions between natural examples and adversarial ones.

++ The experiments are extensive. The evaluation is conducted on both CIFAR and ImageNet under various attacks. The authors also consider the adversarial detection of both white-box and black-box settings. The results are promising and convincing, which achieve better performance than other detectors.

Cons:

-- One of my concerns lies in the computational cost of proposed adversarial detection. According to Figure 2, both a set of natural images and the target image are fed to the pre-trained diffusion model for perturbation generation and expected perturbation score computation, it seems to have a burden of computation compared to other two-sample test methods.

-- It would be better for the authors to provide evaluation of adversarial detection on up-to-date transferable attacks, such as [1,2].

[1]. Enhancing the Transferability of Adversarial Attacks through Variance Tuning. CVPR 2021.

[2]. Improving Adversarial Transferability via Neuron Attribution-based Attacks. CVPR 2022.


**Summary Of The Paper:**

The authors introduce an adversarial detection algorithm which tackles the sensitivity of detection scores from the single sample via proposed expected perturbation score. Specifically, the scores are obtained by the diffusion process which takes into consideration multiple levels of noise. The obtained scores are compared through maximum mean discrepancy. The experiments are conducted on different datasets and settings, which involve the evaluation of different attacks. The proposed adversarial detection algorithm show superiority compared to other baselines.

**Summary Of The Review:**

Overall, I think this paper is interesting and solid. It would be better if the authors can provide more analysis of the inference efficiency and include more evaluation of up-to-date transferable attacks.

---

> ### Author Response · Authors · 2022-11-16
> **Response to Reviewer yyTy**
>
> Thank you for your encouraging comments and detailed suggestions. Responses to the comments are listed below:
>
> **Q1:** One of my concerns lies in the computational cost of proposed adversarial detection. According to Figure 2, both a set of natural images and the target image are fed to the pre-trained diffusion model for perturbation generation and expected perturbation score computation, it seems to have a burden of computation compared to other two-sample test methods.
>
> **A1:** Given a fixed-size model, the computational cost of our method (EPS-AD) mainly depends on two factors: the resolution of input images and total diffusion timestep $T$. Actually, our EPS-AD performs adversarial detection efficiently, especially on low-resolution images, and yields promising performance compared to existing methods. To evaluate the efficiency of EPS-AD, we randomly choose $500$ images from CIFAR-10 and ImageNet respectively in detecting FGSM-L2 adversarial samples on a single RTX3090 GPU. The average time costs per image for CIFAR-10 ($T=20$) and ImageNet ($T=50$) are $0.038s$ and $2.386s$, respectively.
>
> To further demonstrate the effect of total diffusion timesteps on the efficiency, we provide the results under different diffusion timesteps against FGSM-L2 attack, as shown in Table 1 below. From the table, our EPS-AD method shows superior adversarial detection performance when $20 \le T \le 100$ on ImageNet and takes $0.954s$ when $T = 20$, which is much more efficient than that with $T = 50$.  Moreover, our EPS-AD achieves superior or comparable performance on ImageNet and CIFAR-10 compared with existing methods (i.e., KD, LID, and MD) even with $T$ as small as $20$.
>
>
> | AUROC | T=5    | T=10   | T=20   | T=50   | T=100  |  KD  | LID  | MD  |
> |----------|---------|---------|--------|--------|--------|--------|--------|--------|
> | CIFAR-10 | 0.6994 (0.001s) | 0.9566 (0.019s) | 0.9995 (0.038s) | 0.9988 (0.095s) | 0.9985 (0.190s) | 0.9121 | 0.9169 | 0.9995 |
> | ImageNet | 0.5662 (0.239s) | 0.6912 (0.478s) | 0.9917 (0.954s) | 1.0000 (2.386s) | 0.9930 (4.772s) | 0.7004 | 0.8932 | 0.8715 |
>
> Table 1: Adversarial detection performance against FGSM-L2 attack ($\epsilon = 4/255$) under different total diffusion timesteps on CIFAR-10 and ImageNet.
>
> Note that the computational efficiency of our method can be further improved by applying an efficient sampling strategy [A], a low-resolution diffusion model [B] and a sparse diffusion timestep (e.g., sampling with a time interval of $2/1000$ during the diffusion process). We leave these techniques for our future work and will include more discussions about the efficiency issue in our revised version.
>
> [A] DPM-Solver: A Fast ODE Solver for Diffusion Probabilistic Model Sampling in Around 10 Steps. NeurIPS 2022.
>
> [B] Diffusion models beat gans on image synthesis. NeurIPS 2021.
>
>
> **Q2:** It would be better for the authors to provide an evaluation of adversarial detection on up-to-date transferable attacks, such as [1,2].
>
> **A2:** Thank you for your constructive comments. Our proposed EPS-AD can be applied to detecting transferable attacks (e.g., [1, 2]). We have added more evaluation on the transferable attacks, e.g., VMI_FGSM [1] and updated the results below. For another attack method [2], the authors build their code framework on TensorFlow, which needs to be reorganized to compare with our method that is on PyTorch. We have discussed it in related work and will update the comparison results when done.
>
> | Attack Methods | KD | LID | MD | LiBRe | S-N | EPS-N (Ours) | EPS-AD (Ours) |
> |----------|---------|---------|--------|--------|--------|--------|--------|
> | VMI_FGSM | 0.7669 | 0.7386 | 0.8039 | 0.9944 | 0.8638 | 0.9609 | 1.0000 |
>
> Table 2: Comparison of adversarial detection methods against the transferable attack VMI_FGSM ($\epsilon = 4/255$) on ImageNet.

---

> ### Author Response · Authors · 2022-11-21
> **Thanks to reviewer Reviewer yyTy**
>
> Dear Reviewer yyTy,
>
> We have addressed your initial concerns regarding our paper. We are happy to discuss them with you in the openreview system if you feel that there still are some concerns/questions. We also welcome new suggestions/comments from you!
>
> Best regards,
>
> Authors of #575

---

### Official Review · Reviewer_RHkt · 2022-10-24

**Confidence:** 3
**Clarity, Quality, Novelty And Reproducibility:** 1. Within the perturbation process, w…
**Correctness:** 3
**Technical Novelty And Significance:** 4
**Empirical Novelty And Significance:** 3
**Recommendation:** 8

**Strength And Weaknesses:**

Strength:
1. This paper is easy to read.

2. The authors propose a new statistical method, called Expected Perturbation Score (EPS), which is able to obtain enough information to identify adversarial examples with only one example after various perturbations.

3. Sufficient theoretical analysis is performed to demonstrate that EPS is able to simulate the difference between the two distributions under mild conditions. Furthermore, extensive experimental results demonstrate the superiority of the proposed EPS-AD.

4. The proposed EPS will be an effective statistic in many applications, such as out-of-distribution detection and anomaly detection.

Weaknesses:



**Summary Of The Paper:**

The authors propose a new statistic, called Expected Perturbation Score (EPS), for adversarial detection. Based on EPS, the authors develop a Maximum Mean Difference (MMD) metric to measure the difference between test samples and natural samples, and further propose an EPS-based adversarial detection method (EPS-AD). Sufficient theoretical analysis and extensive experiments demonstrate the correctness and effectiveness of the proposed method.

**Summary Of The Review:**

This paper is easy to read and interesting but still has some minor issues, please refer to weaknesses.

---

> ### Author Response · Authors · 2022-11-06
> **Response to Reviewer RHkt**
>
> We thank the reviewer for the encouraging comments and detailed suggestions. Responses are below:
>
> Q1: Within the perturbation process, why do the authors restrict the value of the time-dependent noise schedule t to be in [0, 1000], but only provide results in [0, 100] in Figure 4(f)? Please explain the reason.
>
> A1: In our setting, we set the maximal number of diffusion timestep T = 100, which is sufficient for both EPS-AD and EPS-N to achieve a good solution, and thus it is NOT necessary to use 1000 diffusion timesteps, which can be very time-consuming. Moreover, as the timestep T increases, our methods exhibit progressively better performance, but the performance gain becomes negligible when T is very large. More importantly, according to Theorem 1, a larger diffusion timestep suggests the mean $\mu_S$ in Eq.(9) would gradually approach zero, which will result in a smaller discrepancy between the natural and adversarial distributions. As a result, more diffusion timesteps may deteriorate the detection performance. We will make this clear in our revised version.
>
> Q2: Writing issues.
>
> A2: Thanks for pointing out the typos in our manuscript. We will carefully check the typos and fix them in a revised version.

---

> ### Author Response · Authors · 2022-11-21
> **Thanks to reviewer Reviewer RHkt**
>
> Dear Reviewer RHkt,
>
> We have addressed your initial concerns regarding our paper. We are happy to discuss them with you in the openreview system if you feel that there still are some concerns/questions. We also welcome new suggestions/comments from you!
>
> Best regards,
>
> Authors of #575

---

### Official Review · Reviewer_d5es · 2022-11-08

**Confidence:** 4
**Correctness:** 2
**Technical Novelty And Significance:** 3
**Empirical Novelty And Significance:** 3
**Recommendation:** 5

**Clarity, Quality, Novelty And Reproducibility:**

The paper is well-written and the method is quite simple to understand and implement.

The paper simply combines the existing idea of using score estimation for adversarial perturbation with the min-discrepancy loss -- It's not particularly novel.

**Strength And Weaknesses:**

The main strength of the paper is empirical: the proposed method works very well compared to some of the earlier baselines. The method can effectively detect adversarial perturbation in the low perturbation regime. The author did experiments on both CIFAR data and ImageNet data with different architectures to justify the result.


The main weakness of the paper is three folds: First of all, the proposed method is based on the mean discrepancy. It is not very useful in detecting whether a single image is perturbed or not -- While the previously proposed method can do so for a single image. Hence the paper is only compared to other works in its favorable setting. To that extent, the paper should at least study the dependency of the detection accuracy w.r.t the total number of samples used.

Secondly, the paper also emphasizes that their advantage is "multi-view", where they average the score estimation over time -- I expected the theory to provide some insights on why such averaging is useful, but I could not interpret it from the current form.

Third, I have significant concerns about Definition 1. Here, the author seems to take x_0 = x to define S(x). However, in the "Estimation for expected perturbation score" section, the author claims that they are using s_\theta(x_t, t) to estimate S(x). Note that  s_\theta(x_t, t)  is computed not from x_0 = x, but by averaging over all possible x_0 sampled from distribution p_0. I am not sure this is a good approximation of the S(x) as in Definition 1. The definition probably needs some re-work, but I'm afraid that theorem 1 will be completely falsified after that.








**Summary Of The Paper:**

The paper proposes a new method based on average score function difference to detect adversarial perturbations in a given image. The main result of the paper is empirical, the paper also provides some theoretical justification of the usefulness of the score estimation.

**Summary Of The Review:**

The empirical findings of the paper are solid, although not novel. The mathematics in the paper is a bit concerning and probably needs some rework.

---

> ### Author Response · Authors · 2022-11-18
> **Response to Reviewer d5es [3/3]**
>
> **Q3:** I have significant concerns about Definition 1. Here, the author seems to take $x_0 = x$ to define $S(x)$. However, in the "Estimation for expected perturbation score" section, the author claims that they are using $s_\theta(x_t, t)$ to estimate $S(x)$. Note that $s_\theta(x_t, t)$ is computed not from $x_0 = x$, but by averaging over all possible $x_0$ sampled from distribution $p_0$. I am not sure this is a good approximation of the $S(x)$ as in Definition 1. The definition probably needs some re-work, but I'm afraid that theorem 1 will be completely falsified after that.
>
> **A3:** Using $s_{\theta}(x_{t},t)$  to estimate EPS $S(x)$ (as shown in Section 3.3) is actually reasonable and we explain the detailed reasons as follows. In Definition 1, EPS requires calculating the score $\nabla_{x }\log{p_{t}(x)}$ in each timestep t, where $p_{t}(x)$ is the marginal probability distribution of $x_{t}$ and $x_{0}=x$. Note that the distribution of $x_{t}$ is certainly from the sample $x_{0}$ that can be derived by the given perturbation transition distribution $p_{0t}(x_{t}|x_{0})$. To estimate the score $\nabla_{x }\log{p_{t}(x)}$, following works [r2, r3, r4, r5], we use a neural network $s_{\theta}(x_{t},t)$ to approximate it. The neural network $s_{\theta}(x_{t},t)$ requires training with the samples from the data distribution $p_0$ by optimizing Eq. (14) via score matching. With sufficient data and model capacity, score matching ensures that the optimal solution to Eq. (14), i.e., $s_{\theta}(x_{t},t)$, equals $\nabla_{x }\log{p_{t}(x)}$ for almost all $x$ and $t$ [r3]. As a result, the score $\nabla_{x }\log{p_{t}(x)}$ can be approximated reasonably by $s_{\theta^{*}}(x_{t},t)$. In practice, we use a pre-trained diffusion model to achieve the estimation for the score. We will clarify this in our revised version.
>
> - [r2] Generative modeling by estimating gradients of the data distribution, NeurIPS 2019.
> - [r3] Score-based generative modeling through stochastic differential equations, ICLR 2021.
> - [r4] Variational diffusion models, NeurIPS 2021.
> - [r5] Diffusion Models for Adversarial Purification, ICML 2022.

---

> ### Author Response · Authors · 2022-11-18
> **Response to Reviewer d5es [2/3]**
>
> **Q2:** The paper also emphasizes that their advantage is "multi-view", where they average the score estimation over time -- I expected the theory to provide some insights on why such averaging is useful, but I could not interpret it from the current form.
>
> **A2:** Thanks for your constructive comments. "Multi-view" is an intuitive explanation of why EPS works, i.e., capturing more information from multiple levels of noise perturbations **with one single sample**. In the following, we would like to provide the theoretical analysis and experimental results to clarify the advantages of taking expectation w.r.t. the timestep on the scores.
>
> 1) **Theorem 1 of the paper implies that "multi-view" benefits the stability of distributional discrepancy.** According to Theorem 1, the discrepancy between the natural sample and the adversarial sample is $S(x)-S(\hat{\mathbf{y}}) \stackrel{d}{\rightarrow} \mathcal{N}\left(\mu_{S}, 2 \sigma_{S}^{2} \mathbf{I}\right)$, where $\mu_{S}=E_{t \sim U(0, T)} \mu_t$ with $\mu_{t}=\frac{\epsilon \mathbf{1}}{\gamma_{t}^{2} \sigma_{x}^{2}+\sigma_{t}^{2}}$ and $\sigma_{S}^{2}=E_{t \sim U(0, T)} \sigma_{t}^{2}$ with $\sigma_{t}^{2}=\frac{1}{\gamma_{t}^{2} \sigma_{x}^{2}+\sigma_{t}^{2}}$. Note that $\mu_{t}$ and $\sigma_{t}^{2}$ decrease as the timestep $t$ increases due to the increase of $\gamma_t$ and $\sigma_t$. However, smaller variance $\sigma_{S}^{2}$ and larger means $\mu_{S}$ are required for good adversarial detection. Note that if we only consider one score of some unique timestep $t$ (i.e., removing the expectation from the definition of EPS), the variance $\sigma_{S}^{2}$ and means $\mu_{S}$ of the discrepancy will be so fluctuant that performing detecting adversarial samples will be very sensitive to the timestep $t$ (as validated in Section 4.4). To alleviate this issue, we consider taking expectation w.r.t. the timestep on multiple scores. In this way, the distribution of the discrepancy between the natural sample and the adversarial sample will be more stable to the timestep, which makes it easier to obtain a superior solution.
>
>
> 2) **Experimental results demonstrate that "multi-view" boosts detection performance over various attacks.** Specifically, we set the total timestep $T$ as a hyper-parameter. As shown in Section 4.4, our methods EPS-AD and EPS-N relying on EPS are insensitive to the total timestep $T$, while S-N that only calculates one single score of the sample fluctuates greatly with the timestep. Moreover, as the total timestep $T$ increases, our methods exhibit progressively better performance, but the performance gain becomes negligible or even negative when $T$ is very large. These results coincide with the conclusion in Theorem 1.
>
> We will include the above discussions about this in our revised version.

---

> ### Author Response · Authors · 2022-11-18
> **Response to Reviewer d5es [1/3]**
>
> **Q1:** The proposed method is based on the mean discrepancy. It is not very useful in detecting whether a single image is perturbed or not -- While the previously proposed method can do so for a single image. Hence the paper is only compared to other works in its favorable setting. To that extent, the paper should at least study the dependency of the detection accuracy w.r.t the total number of samples used.
>
> **A1:** We notice that there might be some misunderstandings here. Please note that our method is **not** a batch-based detection method. The main contribution of our paper is the proposal of a new statistic (namely EPS), which is naturally designed for detecting whether a single upcoming example is adversarial. Note that the mean discrepancy is just one way to measure the distance between EPS of an upcoming example and EPSs of natural examples. For example, we can also use the norm of EPSs of examples to find adversarial examples, which is called EPS-N. From our experiments, it is clear that both EPS-AD (EPS with MMD) and EPS-N work better than other methods, verifying that the EPS can help distinguish between natural and adversarial examples.
>
> Specifically, for each sample, its EPS is a vector. If we have $n$ natural samples, we will obtain $n$ EPS vectors for these natural samples. When detecting whether an upcoming sample is adversarial, we can calculate a distance (e.g., MMD) between EPS of the upcoming sample and $n$ EPS vectors of the natural samples. If the EPS of the upcoming sample is far away from such $n$ EPS vectors of the natural samples, we are more confident that the upcoming sample is an adversarial example.
>
> We hope that the above explanations can address your main concern (i.e., if our method is suitable for single sample detection). Since EPS is naturally designed for one example instead of a batch of examples, EPS-based methods (e.g., EPS-AD and EPS-N) can be easily used to detect whether an upcoming sample is adversarial. The main reason is that **an** adversarial example and **a** natural example are already very different in the view of EPS (verified by our experimental results). Additionally, our method EPS-AD is suitable for a different number of natural samples and has superior detection performance with only $100$ natural samples (see the results in Table 13 and Figure 7 in Appendix).
>
> In the following, we will restate our designed MMD that is used to see if one vector (i.e., EPS of an upcoming example in our paper) is close to $n$ vectors (i.e., $n$ EPS vectors of natural examples in our paper). In EPS-AD (i.e., EPS with MMD), we constuct two distributions: $P_{X}=\frac{1}{n}  \sum_{i=1}^{n} \delta (\left \| x -x ^{(i)} \right \| )$ for **a set of natural samples** and $Q_{Y}=\delta (\left \| y -\ \tilde {x} \right \| )$ for **a single test sample**, where $\delta$ is the Dirac function. Then, relying on their EPSs, we calculate the MMD between $P_{X}$ and $Q_{Y}$  as a characteristic of the test sample. Our experiments verify that our MMD can help identify where an upcoming example is adversarial (compared to norm as the distance).

---

> ### Author Response · Authors · 2022-11-18
> **Significance and Novelty of Our Paper**
>
> Thank you for your detailed comments and constructive suggestions. We would like to first clarify the significance of paper and highlight our motivation and main contributions. For convenience of presentation, we will give detailed responses to your three specific concerns: 1) "usefulness of EPS-AD on single-sample detection" in **A1**, 2) "advantages of taking expectation on scores in EPS" in **A2**, and 3) "approximation of EPS" in **A3**, respectively.
>
> **Motivation and Significance of paper.** Deep neural networks (DNNs) are known to be very vulnerable to adversarial samples, which may lead a deep model to make unexpected predictions. In this sense, effectively detecting adversarial samples shall be one critical way to guarantee the trustworthy prediction of DNNs. In this paper, we aim to devise an effective method to detect whether **ONE** given testing sample is adversarial. One direct way is to compare the distributions between adversarial and natural samples. However, this method suffers from two challenges when dealing with **ONE** test sample only：
>
> 1) Existing methods (e.g., Yoon et al. (2021) [r1]) seek to use the score (the gradient of log probability density w.r.t. the sample) of one sample to measure the distributional discrepancy. However, this measure is not accurate enough for detecting one test sample since insufficient information on one single sample makes the distributional discrepancy extremely fluctuant (see discussions on Theorem 1).
>
> 2) Prior works (e.g., Yoon et al. (2021) [r1]) have empirically demonstrated some results about the norm of score in distinguishing adversarial samples from natural samples but did not provide deep investigations or theoretical analysis on it.
>
> To address the above issues,  in this paper we consider obtaining more information of a sample using its "multi-view" perturbations and propose a new statistic called expected perturbation score (EPS) to measure the distributional discrepancy. EPS is essentially the expected score of a sample after various perturbations. Based on this new statistic, we further develop a new adversarial detection method for single-sample adversarial detection. **Our main contributions are as follows:**
>
> 1) **A novel statistic.** We find that the traditional score of one sample is sensitive in identifying adversarial samples due to insufficient information from a single sample only (see Figure 1 and Theorem 1 in the paper). However, we can perturb the sample by adding various noises and thus can obtain various information from its "multi-view" observations or perturbations. We propose to calculate the expected score of the multiple perturbations as the sample's score, namely **expected perturbation score (EPS)**. This new statistic can be much more reliable and effective than traditional score (e.g., Yoon et al. (2021) [r1]) in measuring the distributional discrepancy. Empirically, to calculate the EPS, we propose to use a pre-trained diffusion model to obtain the multi-view scores of a given sample.
>
> 2) **A novel adversarial detection method with extensive empirical justifications.** Based on the novel statistic EPS, we further develop a novel single-sample adversarial detection method called EPS-AD, in which we propose an EPS-based maximum mean discrepancy (MMD) and then use it as a metric to measure the discrepancy between the test sample and natural samples.  We conduct extensive experiments on both CIFAR-10 and ImageNet across various network architectures. Our method consistently outperforms existing methods against $12$ different attacks (known attacks, unseen attacks and transferable attacks), e.g., improving the prior MD method by 1.65%~13.88% AUROC on ImageNet, which verifies the effectiveness of EPS-AD.
>
> 3) **Theoretical justifications.**  First, we provide theoretical analysis that EPS is a proper statistic to distinguish between natural and adversarial data well under mild conditions (see Theorem 1). Specifically, we show that the EPS of the natural sample is closer to those of other natural samples compared to adversarial samples according to their distributions. Second, we theoretically show that the EPS-based MMD between natural and adversarial samples is larger than that among natural samples (see Corollary 1).
>
> [r1] Adversarial purification with score-based generative models. ICML 2021.

---

> ### Author Response · Authors · 2022-11-21
> **Thanks to reviewer Reviewer d5es**
>
> Dear Reviewer d5es,
>
> We have addressed your initial concerns regarding our paper. We are happy to discuss them with you in the openreview system if you feel that there still are some concerns/questions. We also welcome new suggestions/comments from you!
>
> Best regards,
>
> Authors of #575

---

> > ### Author Response · Authors · 2022-12-04
> > **Kind reminder for discussion**
> >
> > Dear  Reviewer d5es,
> >
> > We thank you once again for your valuable comments. We hope you can reassess our work in light of our responses & updates, and reach out with any remaining concerns and questions. We are happy to continue the discussion.
> >
> > Sincerely,
> >
> > Authors of Paper575

---

### Decision · Program_Chairs · 2023-01-20

**Decision:**

Reject

**Justification For Why Not Higher Score:**

The paper has some potential fundamental mathematical flaws in the Lemmas, that can not be easily fixed.

**Justification For Why Not Lower Score:**

N/A

**Metareview: Summary, Strengths And Weaknesses:**

The paper proposes a new method for adversarial detection based on expected perturbation score (EPS), which is computed by applying a pre-trained diffusion model to natural and adversarial images and measuring the difference in their score functions. The paper claims that EPS can better capture the distributional discrepancy between natural and adversarial images than existing methods, and provides some theoretical and empirical support for this claim.

The reviewers have mixed opinions on the paper. Reviewer 1 is very critical of the paper, pointing out several flaws in the definition and estimation of EPS, and questioning the novelty and usefulness of the method. Reviewer 2 is very positive about the paper, praising its clarity, motivation, theoretical analysis, and experimental results. Reviewer 3 is moderately positive about the paper, but raises some concerns about the computational cost and the evaluation of transferable attacks.

After carefully reading the paper and the reviews, we have decided to reject the paper. While we appreciate the authors' attempt to address the sensitivity of adversarial detection to single samples, we find that the paper has several major issues that prevent us from accepting it.

First, we agree with Reviewer 1 that the definition and estimation of EPS are problematic and inconsistent. The paper does not clearly explain how EPS is related to the score function of the diffusion model, and how it is estimated from a single sample. The paper also does not justify why averaging over multiple noise levels is beneficial for adversarial detection, and how it affects the accuracy and efficiency of the method.

Second, we are not convinced by the novelty and significance of the paper. The paper essentially combines two existing ideas: score estimation for adversarial perturbation and maximum mean discrepancy for distributional comparison. The paper does not provide sufficient motivation or analysis for why this combination is novel or superior to other methods. The paper also does not compare with the most relevant and recent baselines, such as those based on out-of-distribution detection or neuron attribution.




**Summary Of Ac-Reviewer Meeting:**

The paper is discussed extensively among the AC, SAC, and PCs. The decision reflects the concern about the correctness of Theorem 1 and a lot of missing conditions in its proof. The AC suggests the authors carefully revise the statement and the proof.